# Identification and characterization of a SARS-CoV-2 specific CD8+ T cell response with immunodominant features

Anastasia Gangaev [1,6], Steven L. C. Ketelaars[1,6], Olga I. Isaeva [1], Sanne Patiwael[1], Anna Dopler [1], Kelly Hoefakker[1], Sara De Biasi [2], Lara Gibellini [2], Cristina Mussini[2], Giovanni Guaraldi [2], Massimo Girardis[2], Cami M. P. Talavera Ormeno[3], Paul J. M. Hekking[3], Neubury M. Lardy[4], Mireille Toebes[1], Robert Balderas [5], Ton N. Schumacher [1], Huib Ovaa [3], Andrea Cossarizza [2] & Pia Kvistborg [1✉]

The COVID-19 pandemic caused by SARS-CoV-2 is a continuous challenge worldwide, and there is an urgent need to map the landscape of immunogenic and immunodominant epitopes recognized by CD8+ T cells. Here, we analyze samples from 31 patients with COVID-19 for CD8+ T cell recognition of 500 peptide-HLA class I complexes, restricted by 10 common HLA alleles. We identify 18 CD8+ T cell recognized SARS-CoV-2 epitopes, including an epitope with immunodominant features derived from ORF1ab and restricted by HLA-A*01:01. In-depth characterization of SARS-CoV-2-specific CD8+ T cell responses of patients with acute critical and severe disease reveals high expression of NKG2A, lack of cytokine production and a gene expression profile inhibiting T cell re-activation and migration while sustaining survival. SARS-CoV-2-specific CD8+ T cell responses are detectable up to 5 months after recovery from critical and severe disease, and these responses convert from dysfunctional effector to functional memory CD8+ T cells during convalescence.

---

[1] Division of Molecular Oncology and Immunology, The Netherlands Cancer Institute, Amsterdam, North Holland, The Netherlands. [2] University of Modena and Reggio Emilia School of Medicine, Modena, Emilia Romagna, Italy. [3] Department of Cell and Chemical Biology, Leiden University Medical Center, Leiden, South Holland, The Netherlands. [4] Department of Immunogenetics, Sanquin Diagnostics B.V., Amsterdam, North Holland, The Netherlands. [5] Department of Biological Sciences, BD Biosciences, San Jose, CA, USA. [6] These authors contributed equally: Anastasia Gangaev, Steven L. C. Ketelaars. This manuscript is dedicated to Huib Ovaa. ✉email: p.kvistborg@nki.nl

The coronavirus disease 2019 (COVID-19) pandemic, caused by the severe acute respiratory syndrome coronavirus 2 (SARS-CoV-2), is an ongoing global emergency. First cases of COVID-19 were reported in December 2019, and as of February 4, 2021, there are more than 104,400,000 confirmed cases and 2,264,000 deaths worldwide[1]. Due to the measures necessary to contain the rapid spread of the infection, this pandemic is having tremendous health and socioeconomic consequences, and there is an urgent need for a better understanding of the natural adaptive immune response.

Accumulating information on the antibody response against SARS-CoV-2 demonstrates that structural proteins, in particular the spike protein and nucleoprotein, serve as potent antibody targets in a substantial fraction of COVID-19 patients[2–7]. However, accumulating data provide evidence for a broader repertoire of immunogenic T cell targets from SARS-CoV-2[4,8–13]. T cell reactivity against SARS-CoV-2 based on peptide pools has been demonstrated by various groups[8–11,13–16], and several studies have identified specific epitopes and their restriction elements[2,4,12,17,18]. However, our understanding regarding which T cell recognized SARS-CoV-2 epitopes are most immunogenic during acute COVID-19 disease is currently limited, as most studies that have identified specific SARS-CoV-2 epitopes are based on convalescent samples[12,17,18]. Knowledge regarding how well the antigens targeted by T cells reflect the antigen composition of current vaccine candidates, and whether these vaccines include immunodominant epitopes, is central for next-generation vaccine development[19,20].

Several studies have characterized the composition of immune cell lineages in COVID-19 patients, which showed that bulk T cells have impaired effector functions (i.e., cytokine production) and higher expression levels of inhibitory receptors compared to bulk T cells from healthy individuals, which is worsening with disease stage[21–23]. However, in-depth characterization of the SARS-CoV-2-specific CD8 T cell response during acute COVID-19 disease directly ex vivo is currently lacking. Such information is essential to improve our understanding about the role of CD8 T cells in the host defense against SARS-CoV-2 and during COVID-19 disease development.

In this work, we probe for CD8 T cell recognition toward 500 SARS-CoV-2 epitope human leukocyte antigen (HLA) complexes, restricted by 10 common HLA class I alleles. A substantial fraction of the identified CD8 T cell responses is directed towards epitopes derived from the open reading frame 1ab polyprotein (ORF1ab), including an epitope with immunodominant characteristics restricted by HLA-A*01:01. In-depth characterization of identified SARS-CoV-2-specific CD8 T cell responses reveals high expression of NKG2A, lack of cytokine production, and a gene expression profile of constrained T cell re-activation. In addition, we show that SARS-CoV-2-specific CD8 T cell responses convert from dysfunctional effector CD8 T cells to functional memory CD8 T cells during convalescence and persist for up to 5 months post COVID-19 disease.

## Results

**Epitope selection**. To cover as many HLA alleles as possible in a patient-specific and high-throughput manner, we focused our analysis on 10 common HLA alleles of the Italian population (samples analyzed in this study were collected from COVID-19 patients in Italy) that could be covered with our peptide-HLA (pHLA) multimer technology. This collection of HLA-A (HLA-A*01:01, HLA-A*02:01, HLA-A*03:01, HLA-A*11:01, and HLA-A*24:02) and HLA-B alleles (HLA-B*07:02, HLA-B*08:01, HLA-B*15:01, HLA-B*18:01, and HLA-B*51:01) resulted in a coverage of approximately 95% of the Italian population (http://www.allelefrequencies.net). For each HLA allele, 50 SARS-CoV-2

epitopes derived from the entire SARS-CoV-2 proteome were selected (Fig. 1a). Epitope selection was based on likelihood of successful proteasomal processing (NetChop-3.1)[24] and predicted binding affinity to HLA (NetMHCpan-4.0)[25]. In addition, SARS-CoV-2 epitopes previously predicted by the science community were included, as well as epitopes shared between SARS-CoV-1 and SARS-CoV-2 for which T cell reactivity has previously been reported (Supplementary Data 1).

**SARS-CoV-2-specific CD8 T cell responses**. To investigate which of the included epitopes from SARS-CoV-2 were recognized by CD8 T cells, peripheral blood mononuclear cell (PBMC) samples from 31 COVID-19 patients with confirmed SARS-CoV-2 infection were analyzed (Table 1 and Supplementary Table 1). Samples of patients with acute critical ($n = 14$), severe ($n = 10$), and moderate ($n = 2$) COVID-19 disease were collected on average 10 days (range −2 to 29) after hospitalization. Samples of asymptomatic patients ($n = 5$) were collected during convalescence (3 months post positive polymerase chain reaction (PCR) test for SARS-CoV-2). SARS-CoV-2-specific CD8 T cell reactivity was evaluated directly ex vivo, using our in-house developed technology based on multiplexing of pHLA multimers conjugated to fluorescent streptavidin reagents. We have previously used this technology to identify tumor-specific CD8 T cell responses in cancer patients[26,27]. A total of 35 SARS-CoV-2-specific CD8 T cell responses specific for 18 different SARS-CoV-2-derived epitopes were detected (Fig. 1b, c and Supplementary Fig. 1). SARS-CoV-2-specific CD8 T cell responses were detected in half of the analyzed patients (17 of 31). The average magnitude of the detected SARS-CoV-2-specific CD8 T cell responses was 1.7% (range: 0.005–19%) of total CD8$^+$ cells (Supplementary Table 2), and the majority (22 of 35) of the responses was restricted by HLA*01:01 (Fig. 1d). CD8 T cell responses specific for epitopes derived from the ORF1ab were of significantly higher magnitude compared to CD8 T cell responses specific for the ORF3a, spike, membrane, and nucleoprotein (Fig. 1e, $P = 0.0309$, two-tailed Mann–Whitney $U$ test). In addition, we analyzed PBMC samples collected prior to November 2019 from 7 healthy donors (HD) covering 7 of the 10 included HLA alleles (HLA-A*01:01, HLA-A*02:01, HLA-A*03:01, HLA-A*11:01, HLA-B*07:02, HLA-B*08:01, and HLA-B*15:01). One CD8 T cell response of low magnitude (0.008% of total CD8$^+$ cells) restricted by HLA-B*15:01 was identified in HD1 (Fig. 1c).

**CD8 T cell recognized SARS-CoV-2 epitopes**. We identified 18 different SARS-CoV-2 epitopes that were recognized by CD8 T cells from COVID-19 patients across 6 of the 10 included HLA alleles (Fig. 1f). Eight of these epitopes were specific for SARS-CoV-2, while 10 epitopes were shared with SARS-CoV-1 and 1 of these epitopes was shared with 3 of the "common cold" coronaviruses (Supplementary Table 3). The origin of these epitopes was diverse, including epitopes derived from the ORF1ab ($n = 8$), spike protein ($n = 6$), membrane protein ($n = 2$), nucleoprotein ($n = 1$), and ORF3a ($n = 1$). Of note, although a high fraction of the selected epitopes was derived from the ORF1ab (driven by the large size of the ORF1ab within the SARS-CoV-2 proteome) we observed an enrichment for CD8 T cell-recognized epitopes derived from the spike protein (Fig. 1g).

**Identified SARS-CoV-2 epitopes are conserved across virus isolates**. Next, we examined whether CD8 T cell-recognized SARS-CoV-2 epitopes were located in positions of the SARS-CoV-2 proteome with a high level of non-synonymous single-nucleotide variants (SNVs). Such information is relevant for the broader utilization of the obtained information. For this

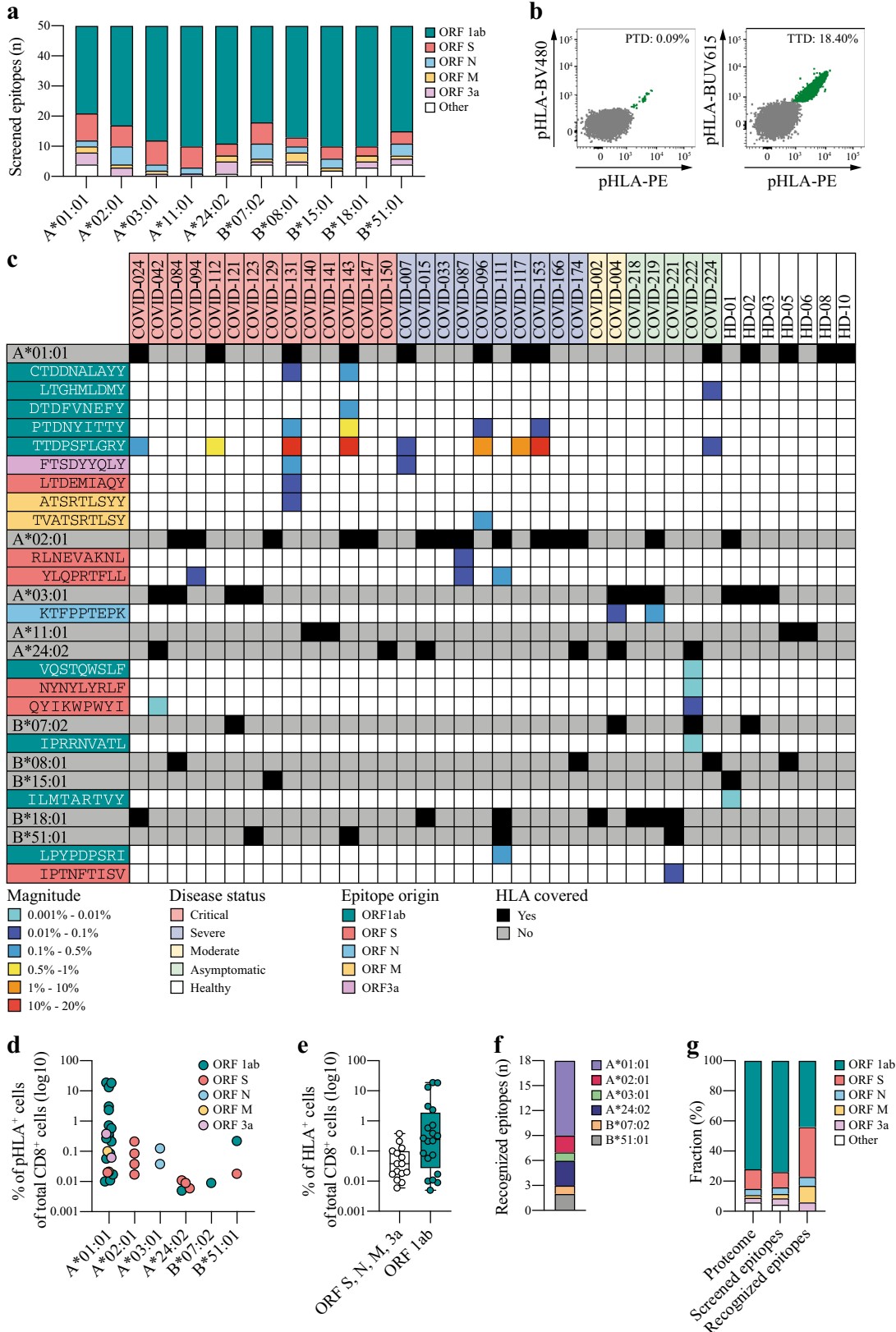

purpose, we obtained information on 385,154 globally isolated SARS-CoV-2 sequences using the COVID-19 CoV Genomics tool[28]. Alternative epitopes with a median SNV frequency of 0.041% (range: 0.01–8.44%) have been reported for 17 of the 18 epitopes that were identified in this study (Supplementary Table 3). Hotspot mutations (SNV frequency cut-off of 2.5% to

exclude background mutations and possible sequencing artifacts) were reported for only 1 (IPTNFTISV) of the 17 epitopes. Interestingly, this hotspot mutation is one of the defining mutations of the recently identified SARS-CoV-2 B117 variant (T716I: IPINFTISV, global alternative SNV frequency 8.44%; https://www.cogconsortium.uk). It is unlikely that the low impact of the T716I

**Fig. 1 Identification of SARS-CoV-2-specific CD8 T cell responses. a** Overview of the selected SARS-CoV-2 epitopes derived from the different SARS-CoV-2 ORFs for each of the 5 HLA-A and 5 HLA-B alleles included in this study. **b** Representative flow cytometric plots of two different SARS-CoV-2-specific CD8 T cell responses detected in patient COVID-153. Magnitude of the response (percentage of double-positive pHLA$^+$ cells [green] of total CD8$^+$ cells [gray]) is indicated. Representative gating strategy is provided in Supplementary Fig. 1. **c** Heatmap of detected SARS-CoV-2-specific CD8 T cell responses ($n = 35$) including information about CD8 T cell-recognized SARS-CoV-2 epitopes ($n = 18$) and their viral protein origin, the magnitude of the response (pHLA$^+$ cells of total CD8$^+$ cells), and HLA coverage as well as the disease status for each COVID-19 patient ($n = 31$) and healthy donor ($n = 7$). Confirmation of identified responses was done for patients COVID-004, 087, 096, 121, 127, 143, and 153 with similar results. **d** Overview of the magnitude, target epitope origin, and number of detected SARS-CoV-2-specific CD8 T cell responses ($n = 35$) across the included HLA alleles. Source data are provided as a Source data file. **e** Difference in the magnitude of SARS-CoV-2-specific CD8 T cell responses targeting epitopes derived from ORF1ab ($n = 20$) compared to CD8 T cell responses targeting epitopes derived from all other SARS-CoV-2 proteins ($n = 15$) combined. All detected responses are included. Box plots indicate the median (line), 25th and 75th percentile (box), min and max (whiskers), and all data points (single circles). Statistical significance was tested with a two-tailed Mann–Whitney $U$ test, $P = 0.0309$. Source data are provided as a Source data file. **f** Number of SARS-CoV-2 epitopes ($n = 18$) that were recognized by CD8 T cells across different HLA alleles. **g** Bar graphs illustrating the contribution of each ORF to the SARS-CoV-2 proteome (left) in comparison to the fraction of the selected epitopes per ORF based on our predictions (middle) and the contribution of each ORF to the CD8 T cell-recognized epitopes (right). ORF open reading frame, S spike, N nucleoprotein, M membrane, pHLA peptide human leukocyte antigen, PTD PTDNYITTY, TTD TTDPSFLGRY.

**Table 1 Characteristics of COVID-19 patients included in the analysis of SARS-CoV-2-specific CD8 T cell responses.**

|  | Critical | Severe | Moderate | Asymptomatic | Healthy |
|---|---|---|---|---|---|
| Total number of patients (n) | 14 | 10 | 2 | 5 | 7 |
| Median age, years (range) | 70 (34 to 77) | 62 (37 to 83) | 64 (36 to 88) | 33 (23 to 51) | 69 (41 to 77) |
| Gender, n (%) |  |  |  |  |  |
| Female | 2 (14%) | 3 (42%) | 0 (0%) | 0 (0%) | 3 (42%) |
| Male | 12 (86%) | 7 (58%) | 2 (100%) | 5 (100%) | 4 (58%) |
| Treatment, n (%) |  |  |  |  |  |
| None | 0 (0%) | 0 (0%) | 2 (100%) | 5 (100%) | 7 (100%) |
| Anakinra | 3 (21%) | 0 (0%) | 0 (0%) | 0 (0%) | 0 (0%) |
| Tocilizumab | 11 (79%) | 10 (100%) | 0 (0%) | 0 (0%) | 0 (0%) |
| Average length of hospitalization, weeks (range) | 5 (2 to 28) | 4 (1 to 16) | 2 (1 to 3) | N/A | N/A |
| Outcome, n (%) |  |  |  |  |  |
| Deceased | 4 (28%) | 1 (10%) | 1 (50%) | 0 (0%) | 0 (0%) |
| Alive | 10 (72%) | 9 (90%) | 1 (50%) | 7 (100%) | 8 (100%) |
| Average time of sample collection, days (range) |  |  |  |  |  |
| After hospitalization | 12 (−2 to 24) | 7 (1 to 29) | 3 (2 to 3) | N/A | N/A |
| After recovery | N/A | N/A | N/A | 12 | N/A |
| HLA coverage, n (%) |  |  |  |  |  |
| Alleles, 1 | 7 (50) | 6 (60%) | 1 (50%) | 0 (0%) | 3 (43%) |
| Alleles, 2 | 5 (35%) | 1 (10%) | 0 (0%) | 4 (80%) | 1 (14%) |
| Alleles, 3 | 2 (15%) | 3 (30%) | 1 (50%) | 1 (20%) | 3 (43%) |
| Number of patients with DR, n (%) | 6 (43%) | 6 (60%) | 1 (50%) | 4 (80%) | 1 (14%) |
| Total number of DR, n | 14 | 12 | 1 | 8 | 1 |
| Median magnitude of DR, % (range) | 2.5% (0.006 to 19%) | 2.059% (0.017 to 18%) | 0.038% | 0.025 (0.005 to 0.13%) | 0.008% |

Detailed information for individual patients is provided in Supplementary Table 1.
*DR* detected SARS-CoV-2-specific CD8 T cell response, *HLA* human leukocyte antigen, *N/A* not available/applicable.

amino acid change will hamper the recognition by T cell receptors (TCRs) specific for IPTNFTISV[29–31]; however, this needs to be experimentally tested in future studies. Overall, our analysis demonstrates that the identified SARS-CoV-2-derived CD8 T cell epitopes are highly conserved across virus variants.

**The ORF1ab encodes an epitope with immunodominant features.** Strikingly, CD8 T cell responses specific for the TTDPSFLGRY (TTD) epitope were detected in all HLA-A*01:01$^+$ patients (8/8) with acute COVID-19 disease (Supplementary Table 3). The magnitude of these CD8 T cell responses was remarkably high: on average >7% of total CD8$^+$ cells (range: 0.074–19%, Fig. 2a and Supplementary Fig. 2). Furthermore, these CD8 T cell responses were of significantly higher magnitude compared to other detected SARS-CoV-2-specific CD8 T cell responses in all COVID-19 patients with acute COVID-19 disease (Fig. 2a, $P = 0.0003$, two-tailed Mann–Whitney $U$ test). Additional SARS-CoV-2-specific CD8 T cell responses were identified for 5 of the 8 HLA-A*01:01$^+$ COVID-19 patients, and the highest magnitude of these responses was on average 70-fold (range: 1.2–216) lower compared to the TTD-specific CD8 T cell response (Fig. 2b, c, $P = 0.0136$, two-tailed Mann–Whitney $U$ test). The

immunodominance hierarchy is partially determined by the naive precursor frequency[32,33]. To further investigate whether more than one T cell clone gave rise to the TTD-specific CD8 T cell response, TCR sequencing data on 542 TTD-specific CD8 T cells obtained from 5 patients was analyzed. In line with the immunodominance pattern based on the presence and the high magnitude of the TTD-specific CD8 T cell responses in all HLA-A*01:01$^+$ patients, we observed a high level of TCR diversity with 150 unique TCR beta-complementarity-determining region 3 (TRB-CDR3) sequences that were expressed in only 1 cell vs. 75 unique TRB-CDR3 sequences that were expressed in ≥2 cells (information for TRB-CDR3 sequences was available for 431/542 cells, Fig. 2d). In addition, we observed an enrichment of TCRs with the TCR beta chain V27 segment within the TTD-specific TCR sequences compared to the TCR repertoire of bulk CD8 T cells (Fig. 2e) suggesting that TCRs specific for the TTD epitope may more likely to be rearranged during T cell development. Together, these data strongly indicate that (depending on the composition of the entire set of HLA alleles) TTDPSFGLGRY is an immunodominant epitope for the subgroup of HLA-A*01:01$^+$ patients, which is present in approximately 30% of the European population (http://www.allelefrequencies.net).

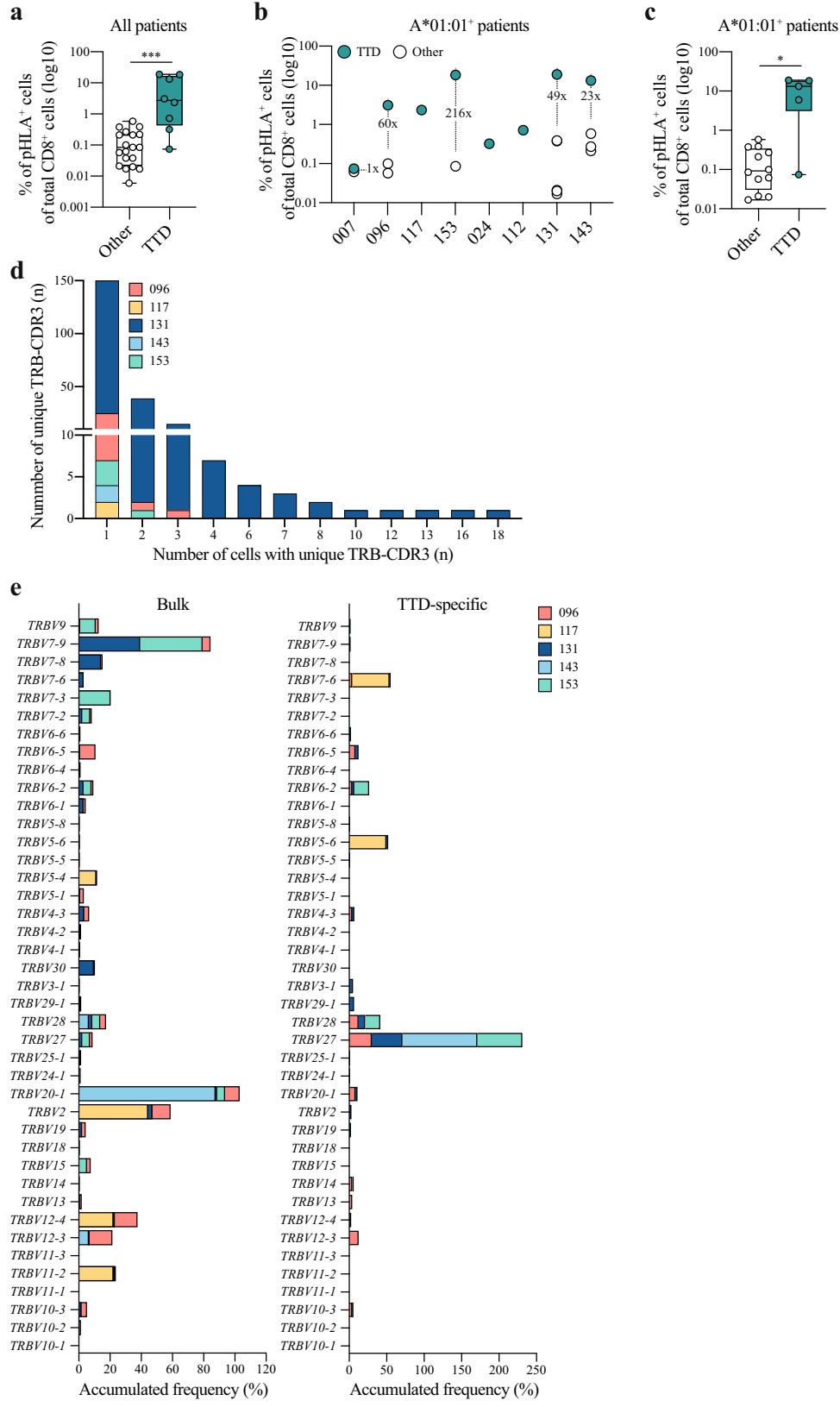

**Dysfunctional state of SARS-CoV-2-specific CD8 T cells**. To characterize the identified SARS-CoV-2-specific CD8 T cell response, we first assessed the functionality based on cytokine production of interferon-γ (IFNγ), tumor necrosis factor (TNF), interleukin (IL)-2, and IL-17 upon stimulation with cognate peptide. The functional assay was focused on the TTD-specific

CD8 T cell responses that were detected in 5 COVID-19 patients with acute COVID-19 disease. The TTD-specific CD8 T cell responses were found to have no (4 patients) or very low levels of cytokine production (1 patient) after 12 h of stimulation (Fig. 3a, b and Supplementary Fig. 3a–c). From one patient, sufficient cell numbers were available to include a cytomegalovirus (CMV)

**Fig. 2 Characterization of the TTD-specific CD8 T cell response. a** Difference in the magnitude of SARS-CoV-2-specific CD8 T cell responses specific for the TTD epitope ($n = 8$) compared to CD8 T cell responses specific for all other epitopes ($n = 27$). Responses included in this analysis were detected in COVID-19 patients with acute disease. Box plots indicate the median (line), 25th and 75th percentile (box), min and max (whiskers), and all data points (single circles). Statistical significance was tested with a two-tailed Mann–Whitney U test, $P = 0.0003$. Source data are provided as a Source data file. **b** SARS-CoV-2-specific CD8 T cell responses specific for the TTD epitope (green) and other epitopes (white) detected in HLA-A*01:01+ patients with acute critical and severe disease ($n = 8$). The fold difference in magnitude between the TTD-specific CD8 T cell response and the second-highest detected SARS-CoV-2-specific CD8 T cell response is shown if present. **c** Difference in the magnitude of TTD-specific CD8 T cell responses ($n = 5$) compared to responses specific for other HLA-A*01:01-restricted epitopes ($n = 12$) that were detected in HLA-A*01:01+ patients. Box plots indicate the median (line), 25th and 75th percentile (box), min and max (whiskers), and all data points (single circles). Statistical significance was tested with a two-tailed Mann–Whitney U test, $P = 0.0136$. Source data are provided as a Source data file. **d** Quantity of unique TRB-CDR3 chains present in TTD-specific CD8 T cells ($n = 431$) of COVID-19 patients ($n = 5$). The number of single cells expressing unique TRB-CDR3 chains is indicated on the x-axis. TRB-CDR3 T cell receptor beta-complementarity-determining region 3. **e** Frequency of cells expressing different TRBV segments that were found in bulk CD8 T cells (left panel, $n = 1860$) compared to TTD-specific CD8 T cells (right panel, $n = 431$) of COVID-19 patients ($n = 5$). Accumulated (stacked) frequencies for each individual patient are shown. TTD TTDPSFLGRY, pHLA peptide human leukocyte antigen. *** $P < 0.001$, * $P < 0.05$.

epitope, and CD8 T cells reactive toward this epitope were found to be functional (Supplementary Fig. 3d, e). Of note, the technical control using phorbol 12-myristate 13-acetate and Ionomycin led to the production of IFNγ, TNF, and IL-2 (Supplementary Fig. 3f). Importantly, these results demonstrate that detection of these responses with a functional readout failed or vastly underestimated the magnitude of the TTD-specific CD8 T cell responses during acute disease. To further examine the SARS-CoV-2-specific CD8 T cells, we measured the expression levels (ex vivo) of the inhibitory receptors NKG2A and programmed cell death protein 1 (PD-1) on SARS-CoV-2-specific CD8 T cells compared to bulk CD8 T cells from COVID-19 patients with acute disease. The fraction of NKG2A+ cells was significantly higher for SARS-CoV-2-specific CD8 T cells compared to bulk CD8 T cells but not for PD-1 (Fig. 3c, d, NKG2A: $P = 0.0427$, PD-1: $P = 0.7651$, two-tailed Mann–Whitney U test). In addition, we observed a positive correlation between the magnitude of the response and the expression levels (mean fluorescence intensity (MFI)) of NKG2A and PD-1 (Fig. 3e, NKG2A: $r = 0.5$, $P = 0.0199$, PD-1: $r = 0.5$, $P = 0.0361$, Spearman correlation analysis). Together, these data suggest that the identified TTD-specific CD8 T cell responses in patients with ongoing severe and critical COVID-19 disease were highly dysfunctional and that SARS-CoV-2-specific CD8 T cell responses displayed a high level of activation/inhibitory receptors.

**Transcriptomics show constrained T cell re-activation profile.** To further characterize the state of the TTD-specific CD8 T cell responses during acute COVID-19 disease, we performed single-cell RNA sequencing on bulk and TTD-specific CD8 T cells from COVID-19 patients with acute severe and critical disease using the 10× Genomics Chromium system (Supplementary Fig. 4a). We obtained data on a total of 3064 CD8 T cells from 6 patients with acute COVID-19 disease (Table 2). The obtained cells were sorted and sequenced in two independent experiments (batch I and batch II). In the first experiment, we obtained data on bulk CD8 T cells from 5 patients (batch I: COVID-087, 096, 117, 143, 153, Supplementary Fig. 4b) and on TTD-specific CD8 T cells from 4 of these patients (COVID-087 was negative for HLA-A*01:01 and did therefore not harbor TTD-specific CD8 T cells). In the second experiment, we obtained data from 1 additional patient including bulk and TTD-specific CD8 T cells (batch II: COVID-131, Supplementary Fig. 4b). Louvain clustering based on highly variable genes resulted in six and eight clusters in batch I and batch II, respectively (Fig. 4a and Supplementary Data 2). We compiled a consensus list of well-established T cell differentiation gene markers (Supplementary Data 3), which allowed us to discriminate naive (batch I and batch II: C1) from effector (batch I: C2–C6 and batch II: C2–C7) and memory CD8 T cells

(C8 only clearly separated in batch II, Fig. 4b). Data on 542 TTD-specific CD8 T cells was obtained for 5 of the 6 patients (Table 2), with the vast majority of cells derived from 2 patients (batch I: COVID-096 and batch II: COVID-131). TTD-specific CD8 T cells were predominantly present in cluster C2, C4, and C6 in batch I, while TTD-specific CD8 T cells in batch II formed a separate cluster (C2, Fig. 4c). This difference in clustering is likely based on the difference in numbers of TTD-specific CD8 T cells (batch I: $n = 48$, batch II: $n = 494$). Subsequently, we performed differential gene expression analysis between TTD-specific CD8 T cells and bulk CD8 T cells from the naive cluster (C1). The analysis resulted in 175 and 2594 differentially expressed genes in TTD-specific CD8 T cells in batch I and batch II, respectively (Supplementary Fig. 4c and Supplementary Data 4). Gene ontology analysis revealed the presence of several immune response-related processes, including T cell activation, regulation of T cell activation, adhesion, and proliferation in the TTD-specific CD8 T cells compared to the naive cells (Supplementary Data 4). Next, we assessed the difference in gene expression between TTD-specific CD8 T cells and the effector/memory bulk CD8 T cells (C2–C8). We focused our analysis on genes that were identified in both data sets to be either upregulated ($n = 12$) or downregulated ($n = 209$) in TTD-specific CD8 T cells compared to bulk effector/memory CD8 T cells (Supplementary Fig. 4d and Supplementary Data 5). Upregulated genes were associated with T cell survival (e.g., *CD27*) and TCR signaling (e.g., *PAG1* and *COTL1*) while downregulated genes were associated with T cell activation (e.g., *TNFRSF9* [CD137] and *LMNA*), migration to epithelial tissue and site of inflammation (e.g., *CXCR2*, *CCL3*), proliferation (e.g., *MYBL1*, *CDK2*, and *USP37*), and effector function (e.g., *GNLY*). Together, these results suggest that activated TTD-specific CD8 T cells display a gene expression program of maintained cell survival but restricted T cell (re) activation, proliferation, and migration, which is in line with the lack of ability to produce cytokines in response to peptide stimulation.

**SARS-CoV-2-specific CD8 T cells convert to functional memory cells.** To investigate whether SARS-CoV-2-specific CD8 T cell responses persist after recovery from acute COVID-19 disease, we first analyzed the kinetics of the identified CD8 T cell responses from 3 COVID-19 patients during acute disease and convalescence (4–5 months post hospital discharge). All of the six previously identified SARS-CoV-2-specific CD8 T cell responses were detected during convalescence (Fig. 5a, b). The magnitude of the SARS-CoV-2-specific CD8 T cell responses ($n = 5$) detected in COVID-96 and -143 was on average 11-fold lower during convalescence compared to acute COVID-19 disease. Nevertheless, the magnitude of the TTD-specific CD8 T cell response in

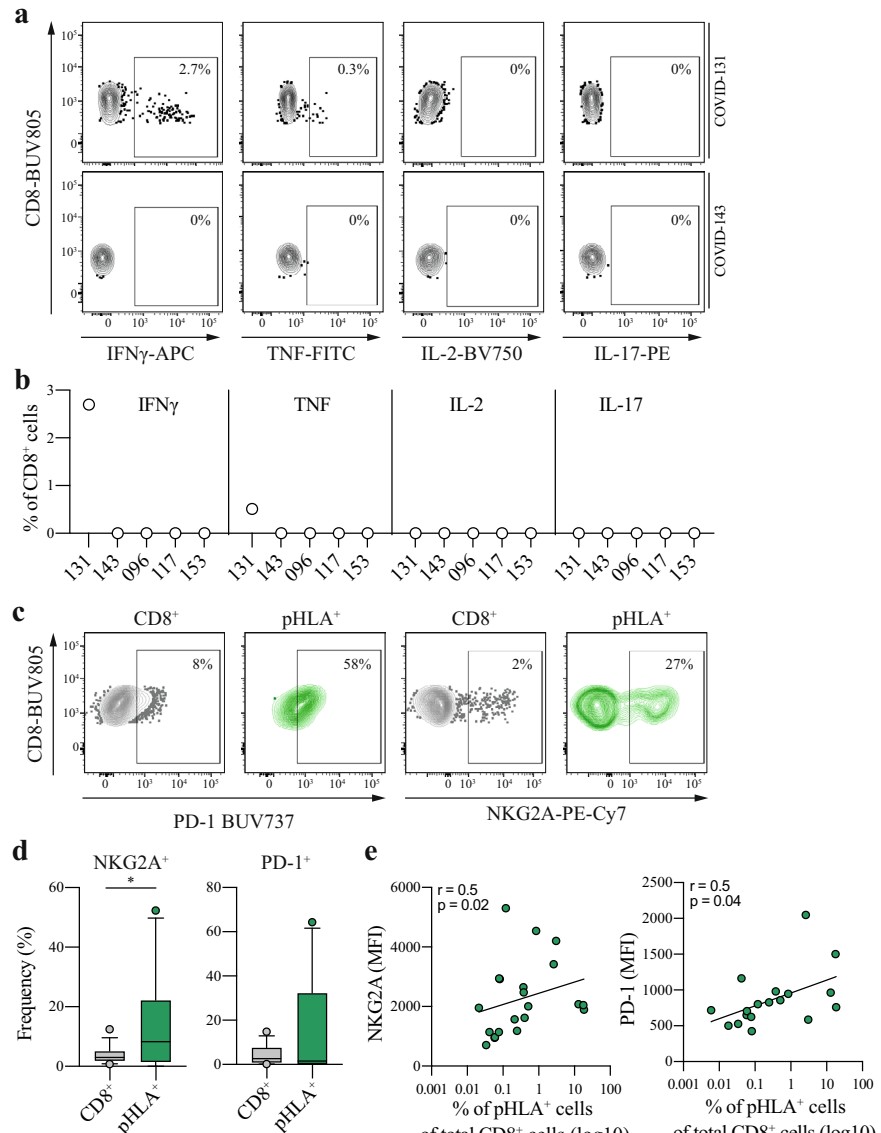

**Fig. 3 Characterization and functional assessment of SARS-CoV-2-specific CD8 T cells. a** Representative gating used to assess the functional capacity of TTD-specific CD8 T cells of 2 COVID-19 patients. Functionality was assessed based on production of IFNγ, TNF, IL-2, and IL-17 after 12 h peptide stimulation with TTD. Percentages represent the frequency of cytokine-producing cells. Full gating strategy is shown in Supplementary Fig. 3a–c. **b** Expression of IFNγ, TNF, IL-2, and IL-17 in CD8 T cells of COVID-19 patients ($n = 5$) with acute disease after 12 h peptide stimulation with TTD. DMSO control-subtracted percentages are shown. Two independent experiments were performed for patient COVID-143 and COVID-153 with similar results. **c** Representative flow cytometric plots illustrating the gating strategy used to quantify the fraction and expression levels of PD-1 and NKG2A on total CD8 T cells (CD8+, gray) and SARS-CoV-2-specific CD8 T cells (pHLA+, green) in COVID-112. The gates applied for detected SARS-CoV-2-specific CD8 T cell responses were based on total CD8 T cells. **d** Differences in the fraction of NKG2A+ or PD-1+ cells of SARS-CoV-2-specific CD8 T cells (pHLA+, $n = 27$) compared to bulk CD8 T cells (CD8+, $n = 32$) of COVID-19 patients with acute disease. Box plots indicate the median (line), 25th and 75th percentile (box), 5th and 95th percentile (whiskers), and outliers (single circles). Statistical significance was tested with a two-tailed Mann–Whitney $U$ test, NKG2A: $P = 0.0427$, PD-1: $P = 0.7651$. Source data are provided as a Source data file. **e** Correlation analysis between the expression levels (MFI) of NKG2A+ (pHLA+, $n = 21$) or PD-1+ (pHLA+, $n = 18$) on SARS-CoV-2-specific CD8 T cells and the magnitude of SARS-CoV-2-specific CD8 T cell responses. Two-tailed Spearman correlation analysis was performed, NKG2A: $P = 0.0199$, $r = 0.5$, PD-1: $P = 0.0361$, $r = 0.5$. TTD TTDPSFLGRY, pHLA peptide human leukocyte antigen. * $P < 0.05$. Source data are provided as a Source data file.

COVID-143 remained considerably high (>1% of total CD8 T cells) during convalescence. For patient COVID-117, the magnitude of the TTD-specific CD8 T cell response was unaltered during convalescence. Phenotypic characterization of these responses by flow cytometry revealed that the percentage of SARS-CoV-2-specific CD8 T cells expressing activation and effector T cell markers (HLA-DR, CD95, and CXCR3) was significantly decreased. In contrast, the percentage of SARS-CoV-2-specific CD8 T cells expressing the T cell differentiation markers

CD45RA and CCR7 (albeit at a lower level compared to bulk naive T cells) was significantly increased during convalescence (Fig. 5c, d and Supplementary Fig. 5c). In addition, SARS-CoV-2-specific CD8 T cells displayed lower expression levels (MFI) of PD-1 during convalescence compared to acute disease (Fig. 5e, f). Lastly, we assessed the functionality of SARS-CoV-2-specific CD8 T cell responses during acute disease and convalescence. Sufficient material for functional assessment was available for two of the three patients. In contrast to absent cytokine production

**Table 2 Number of CD8 T cells and TTD-specific CD8 T cells obtained from single-cell RNA-sequencing per patient.**

| Batch | Patient | CD8 T cells (*n*) | CD8 T cells with TCR (*n*) | TTD-specific CD8 T cells (*n*) | TTD-specific cells with TCR (*n*) |
|---|---|---|---|---|---|
| Batch I | COVID-096 | 1074 | 799 | 33 | 23 |
| Batch I | COVID-087 | 35 | 21 | 0 | 0 |
| Batch I | COVID-153 | 33 | 25 | 8 | 5 |
| Batch I | COVID-143 | 26 | 18 | 5 | 2 |
| Batch I | COVID-117 | 12 | 11 | 2 | 2 |
| Batch II | COVID-131 | 1884 | 1479 | 494 | 407 |

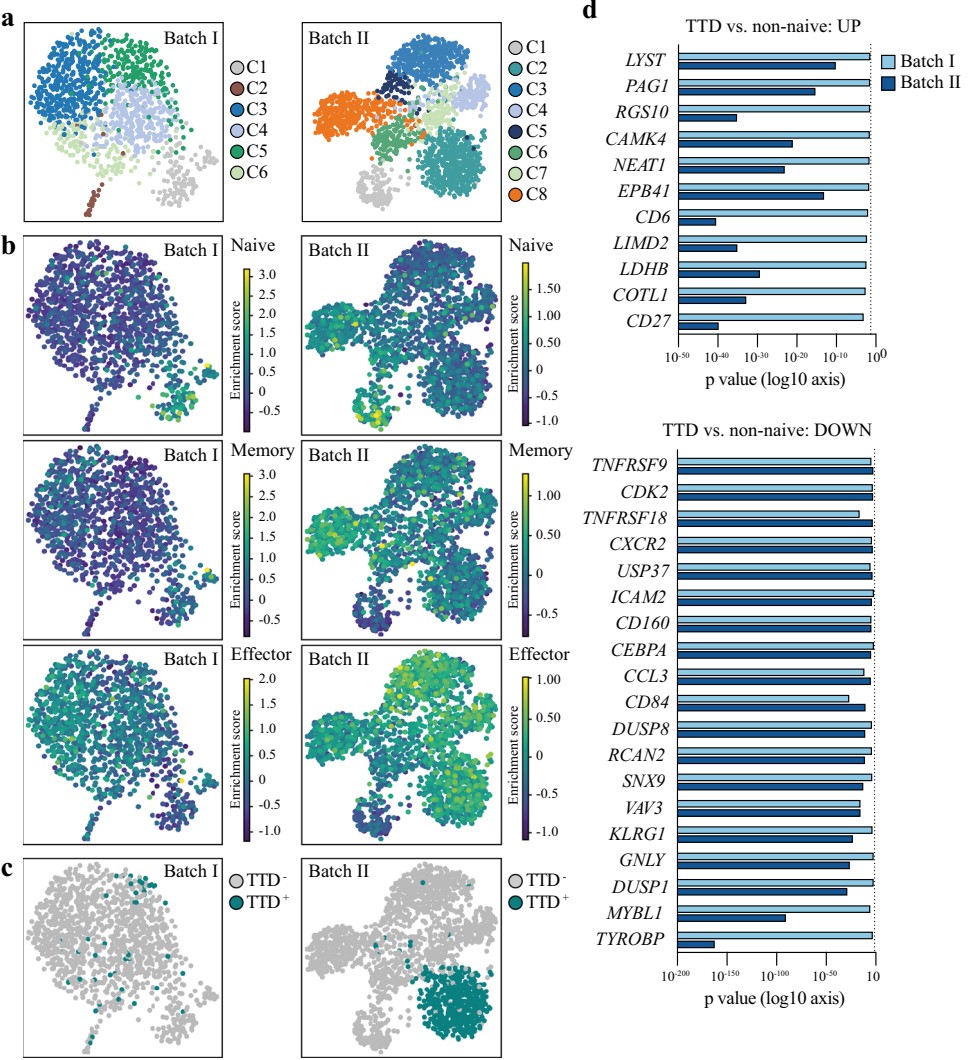

**Fig. 4 Single-cell transcriptome analysis of SARS-CoV-2-specific CD8 T cells. a** UMAP representations of cell clusters (batch I: *n* = 6 and batch II: *n* = 8) of the single-cell gene expression data of CD8 T cells (batch I: *n* = 1180 and batch II: *n* = 1884) isolated from COVID-19 patients (batch I: *n* = 5 and batch II: *n* = 1). Clusters were identified using the Louvain algorithm. Full list of differentially expressed genes for each cluster is provided in Supplementary Data 2. **b** UMAP representations of enriched gene signatures (Supplementary Data 3) used to identify naive, memory, and effector CD8 T cells in batch I and batch II. **c** UMAP representations of single-cell gene expression data of TTD-specific CD8 T cells (batch I: *n* = 48 and batch II: *n* = 494) isolated from COVID-19 patients (batch I: *n* = 4 and batch II: *n* = 1). **d** Differentially expressed genes that were found to be significantly (FDR < 0.05) upregulated (ln fold change of >1) or downregulated (ln fold change <−1) in TTD-specific CD8 T cell compared to non-naive CD8 T cells in batch I and batch II. The full list of genes is provided in Supplementary Data 5. Statistical significance was tested with a two-tailed *t* test; multiple correction was performed with the Benjamini–Hochberg procedure. Original *P* values are displayed on a log10 *x*-axis. TTD TTDPSFLGRY.

during acute disease, TTD-specific CD8 T cells from both patients gained the capacity to secrete IFNγ and TNF during convalescence (Fig. 5g, h). Together, these results demonstrate that SARS-CoV-2-specific CD8 T cell responses can be detected

up to 5 months post recovery from acute severe and critical COVID-19 disease and that these cells convert from dysfunctional effector CD8 T cells to functional memory CD8 T cells during convalescence.

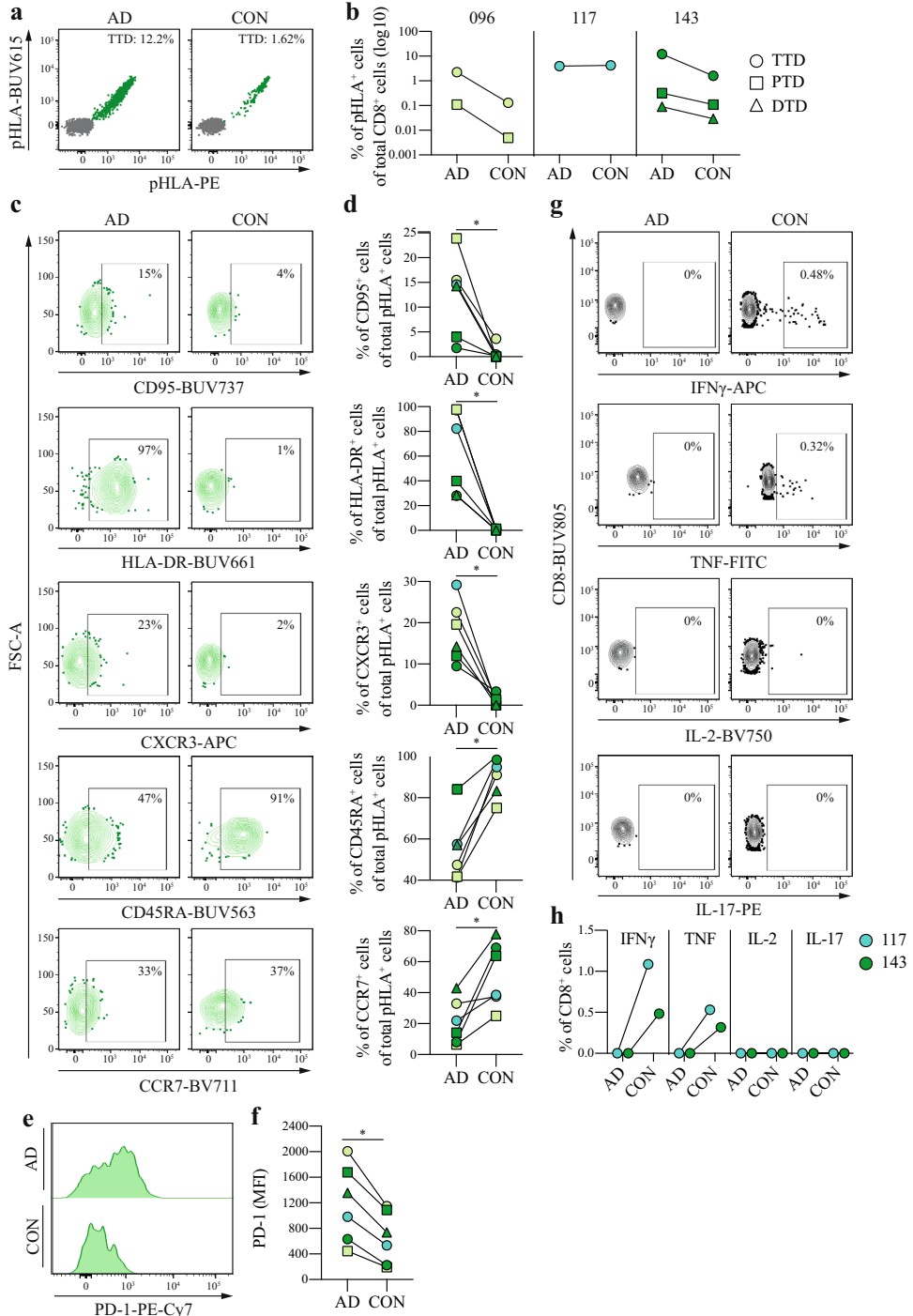

## Discussion

We employed a systematic effort to identify CD8 T cell-recognized SARS-CoV-2 epitopes by covering 10 common HLA alleles and a selection of in silico predicted peptides from the entire SARS-CoV-2 proteome resulting in 500 unique pHLA complexes. Our analyses resulted in the identification of 18 SARS-CoV-2-derived CD8 T cell epitopes. The origin of these epitopes was diverse, including the ORF1ab, spike protein, membrane protein, nucleoprotein, and the ORF3a. Eight of the 18 identified epitopes were derived from the ORF1ab including one epitope (TTDPSFLGRY), which displayed strong immunodominant characteristics. First, CD8 T cell responses specific for this epitope were detected in all HLA-A*01:01+ patients with acute

COVID-19 disease. Second, the magnitude of TTD-specific CD8 T cell responses was remarkably high (average: 7%, range: 0.074–19%) and substantially higher compared to the other SARS-CoV-2-specific CD8 T cell responses across and within most patients. Notably, such remarkably high magnitude of SARS-CoV-2-specific CD8 T cell responses has not been reported in previous studies[2,4,9–14,16–18,34]. Third, TCR sequencing data of the TTD-specific CD8 T cell responses revealed a high level of clonal diversity based on TRB-CDR3 regions, together with an enrichment for the TRBV27 segment suggesting that these TCR rearrangements may occur more frequently similar to what is known for, e.g., TCRs specific for the immunodominant CMV epitope NLVPMVATV[35,36]. These data provide the first TCR

**Fig. 5 Characterization of SARS-CoV-2-specific CD8 T cells during acute disease and convalescence. a** Representative dot plots of the TTD-specific CD8 T cell response detected in patient COVID-143. Magnitude of the response (percentage of double-positive pHLA+ cells [green] of total CD8+ cells [gray]) is indicated. Representative gating strategy is shown in Supplementary Fig. 5. **b** Magnitude of SARS-CoV-2-specific CD8 T cell responses (n = 6) detected in COVID-19 patients (n = 3). Source data are provided as a Source data file. **c** Representative gating strategy used to quantify the fraction of CD95+, CXCR3+, HLA-DR+, CD45RA+, and CCR7+ TTD-specific CD8 T cells in COVID-096. Gates were set based on bulk CD8 T cells (representative gating strategy is shown in Supplementary Fig. 5). **d** Fraction of CD95+, CXCR3+, HLA-DR+, CD45RA+, and CCR7+ SARS-CoV-2-specific CD8 T cell responses (n = 6) detected in COVID-19 patients (n = 3). Individual points are annotated according to CD8 T cell specificity for individual patients as shown in Fig. 5b. Statistical significance was tested with a two-tailed Wilcoxon signed-rank test, P = 0.0312. Source data are provided as a Source data file. **e** Representative histogram of the expression levels (MFI) of PD-1 on SARS-CoV-2-specific CD8 T cells in COVID-096. **f** Expression levels (MFI) of PD-1 on SARS-CoV-2-specific CD8 T cell responses (n = 6) detected in COVID-19 patients (n = 3). Individual points are annotated according to CD8 T cell specificity for individual patients as shown in Fig. 5b. Statistical significance was tested with a two-tailed Wilcoxon signed-rank test, P = 0.0312. Source data are provided as a Source data file. **g** Representative gating used to assess the expression of IFNγ, TNF, IL-2, and IL-17 in CD8 T cells in patient COVID-143 after 12-h peptide stimulation with TTD. Percentages of cytokine-producing cells are indicated. Full gating strategy is shown in Supplementary Fig. 3a–c. **h** Functionality of CD8 T cells (n = 2) after 12-h peptide stimulation with TTD. DMSO control-subtracted percentages are shown. Two independent experiments were performed for patient COVID-143 with similar results. AD acute disease, CON convalescence, TTD TTDPSFLGRY, PTD PTDNYITTY, DTD DTDFVNEFY, pHLA peptide human leukocyte antigen. * P < 0.05.

sequences specific for an immunodominant SARS-CoV-2 epitope. Such information can pave the way for T cell therapies in COVID-19 patients unable to mount protective T cell immunity in response to, e.g., vaccination.

Although our analysis was limited by the selection of HLA alleles and the number of epitopes, as well as imperfect in silico epitope predictions, accumulating evidence from other groups[8,12,18] and our data demonstrate that epitopes derived from other parts of the SARS-CoV-2 proteome than the spike protein can induce high-magnitude CD8 T cell responses. It is currently not known whether any of the epitopes derived from the spike protein have immunodominant properties in the absence of, e.g., the ORF1ab, and it remains to be established whether CD8 T cell responses raised against the spike protein alone are sufficient to induce long-lasting protective T cell immunity. The spike protein is the focus of the currently approved vaccines against SARS-CoV-2[37,38], and future studies assessing vaccine-induced T cell responses will be able to answer such outstanding questions.

A number of recent studies provide evidence suggesting that individual genetic variation in HLA type may predict COVID-19 outcome across a population[39–41]. Such a scenario is well described for, e.g., HLA-B*57:01+ HIV patients who are more likely to be long-term non-progressors[42]. One study on COVID-19 patients showed that HLA-A*01:01 and HLA-A*02:01 were associated with higher risk for severe disease in Italy; however, this analysis was not based on separate HLA alleles but rather combinations[40]. Based on our data, it appears that there may be a bias for HLA alleles capable of presenting epitopes, which can induce CD8 T cell responses of profound magnitude. We detected 27 SARS-CoV-2-specific CD8 T cell responses in samples collected during acute COVID-19 disease. The average magnitude of the 20 HLA-A*01:01-restricted CD8 T cell responses that were detected in patients with acute COVID-19 disease was 2.97% (range: 0.017–19%) of total CD8 T cells. In contrast, the average magnitude of the 7 detected CD8 T cell responses restricted by other HLA alleles was 0.084% (range: 0.006–0.22%) of total CD8 T cells. This difference in magnitude was not only driven by CD8 T cell responses specific for the TTD epitope, as the average magnitude of the 12 other (not TTD-specific) HLA-A*01:01-restricted CD8 T cell responses was 2-fold higher compared to CD8 T cell responses restricted by other HLA alleles (0.182 vs. 0.082%). This observation suggests that patients with a specific (set of) HLA allele(s) may be able to raise a substantial CD8 T cell response toward SARS-CoV-2. However, given the limited size of patients in our and other studies[39–41], this hypothesis needs to be addressed in larger patient cohorts in the future.

Interestingly, in-depth characterization of SARS-CoV-2-specific CD8 T cells based on a stimulation assay, ex vivo flow cytometry, and transcriptome analysis revealed a regulated activation program that maintains CD8 T cell survival while limiting their effector function during acute COVID-19 disease. First, we observed a lack or strong reduction of cytokine production (IFNγ, TNF, IL-2, and IL-17) upon stimulation with the TTD epitope demonstrating a highly dysfunctional state of SARS-CoV-2-specific CD8 T cells during acute severe and critical COVID-19 disease. These data are in line with other studies demonstrating that the SARS-CoV-2-specific CD8 T cell response is functionally impaired in COVID-19 patients with severe disease in comparison to patients with acute mild and moderate disease[9,11,43,44]. Second, the percentage of SARS-CoV-2-specific CD8 T cells expressing NKG2A was significantly higher compared to bulk CD8 T cells, and the magnitude of SARS-CoV-2-specific CD8 T cell responses correlated with increased expression of NKG2A and PD-1. The expression of the NKG2A receptor, in particular, may in part explain the inability of SARS-CoV-2-specific CD8 T cells to produce cytokines. Expression of NKG2A can limit immunopathology in influenza by inhibiting effector function[45]. This receptor may have a similar role in COVID-19 patients who frequently present with abnormally high levels of IL-6[21,46–48], which drives the expression of the NKG2A ligand HLA-E on T cells, B cells, and macrophages[49,50]. Third, several studies showed that bulk T cells have impaired effector functions (i.e., cytokine production) and higher expression levels of inhibitory receptors compared to bulk T cells from healthy individuals, which is worsening with disease stage[21–23]. In line with the functional data, our results demonstrate that identified SARS-CoV-2-specific CD8 T cell responses harbor a regulated activation program that limits their functional capacity and migration in patients with acute severe and critical COVID-19 disease. These data may be a cautious suggestion that SARS-CoV-2-specific CD8 T cells—unlike CD4 T cells[51]—are less likely to contribute to the immunopathology observed in patients with severe and critical COVID-19[52] but rather have a protective role, which is in line with findings from other studies[43,44].

Our longitudinal data analysis demonstrates that SARS-CoV-2-specific CD8 T cell responses persist up to 5 months post hospital discharge (as expected at a lower magnitude compared to acute disease for most responses) and that these responses convert to functional memory T cells. These findings are in line with previous studies demonstrating functional memory SARS-CoV-2-specific CD8 T cell responses at the time of convalescence[8,10,12,14,18]. Such memory T cell responses can persist up to 6 years in recovered SARS-CoV-1 patients[53], and it is of key importance to understand if

this will also be the case for the SARS-CoV-2-specific T cell response.

To summarize, we have identified a SARS-CoV-2 epitope with immunodominant features, which is encoded by the ORF1ab and restricted by a common HLA allele. Interestingly, our data provide evidence for a dysfunctional state of SARS-CoV-2-specific CD8 T cell responses during acute infection and that these responses convert to functional memory T cells during convalescence.

## Methods

**Patient material.** Peripheral blood samples of COVID-19 patients and HDs were collected in accordance with the Declaration of Helsinki after approval by the institutional review boards (Ethical Committee of Area Vasta Emilia Romagna, protocol number 177/2020, March 10, 2020, and subsequent amendments). Each participant gave informed consent. All COVID-19 patients were tested positive for SARS-CoV-2 using reverse transcriptase PCR from an upper respiratory tract (nose/throat) swab test in accredited laboratories. Patients were assigned in groups based on the COVID-19 disease status during sample collection according to the World Health Organization guidelines[54]. Samples were collected as follows: during hospitalization/acute disease for COVID-19 patients with moderate, severe, or critical disease, 3 months post positive PCR test for asymptomatic COVID-19 patients, before November 2019 for HDs, and 4–5 months post hospital discharge, and with a negative SARS-CoV-2 PCR test, for convalescent patients who recovered from severe or critical COVID-19 disease (Table 1 and Supplementary Table 1). Peripheral blood was collected in ethylenediaminetetraacetic acid tubes following subsequent isolation of PBMCs using Ficoll-Paque density centrifugation according to standard protocol. PBMCs were suspended in fetal bovine serum (FBS, Sigma, F7524) with dimethyl sulfoxide (DMSO, Sigma, D4540, 10% v/v) and stored in liquid nitrogen.

**HLA typing.** PBMCs isolated from COVID-19 patients were thawed and washed with RPMI 1640 (Life Technologies, 21875-034) supplemented with FBS (Sigma, F7524, 10% v/v), penicillin–streptomycin (Life Technologies, 15140-122, 1% v/v), and benzonase nuclease (Merck-Millipore, 70746-4, 2500 U/mL), resuspended and incubated at 37 °C for 30 min. For the HD samples, DNA was isolated direct from whole blood. PBMCs were counted and up to 1,000,000 cells were aliquoted for subsequent DNA isolation. DNA was isolated using the DNeasy Blood & Tissue Kit (Qiagen, 69506) according to the manufacturer's protocol (Qiagen). HLA typing was done using next-generation sequencing according to the manufacturer's protocol (GenDx).

**SARS-CoV-2 epitopes.** The entire SARS-CoV-2 proteome obtained from UniProt (Proteome ID: UP000464024) was considered for predicting SARS-CoV-2 epitopes using R version (v.3.6.3) and data.table (v.12.6). List of selected SARS-CoV-2-epitopes is provided in Supplementary Data 1. Potential SARS-CoV-2 epitopes were first selected based on a predicted NetChop-3.1[24] proteasomal processing score >0.5, followed by a final selection step based on the highest predicted binding affinity to major histocompatibility complex (MHC) according to NetMHCpan-4.0[25]. In addition, SARS-CoV-2 epitopes that were predicted to be most immunogenic by the science community were included for analysis[34,55–57]. Predicted 9-11mer epitopes from the following ORFs of the SARS-CoV-2 proteome were included in the final analysis: ORF 1ab, 3a, 6, 7a, 7b, 8, 9b, 10, 14, envelope, membrane, nucleoprotein, and spike protein. Fifty SARS-CoV-2 epitope–HLA combinations were selected for each of the top 10 most prevalent HLA alleles in Italy, which resulted in a total of 438 unique peptides that were synthesized by the Chemical Biology group, Leiden University Medical Centre. Predicted scores for proteasomal processing (NetChop-3.1) and binding affinity (NetMHCpan-4.0) of the 50 selected and 18 CD8 T cell-recognized epitopes for each allele are shown in Supplementary Fig. 6. The COVID-19 CoV Genomics tool (v.1.6.0, data cut-off: 28-01-2021)[28], enabled by the GISAID initiative, was used to determine whether CD8 T cell-recognized epitopes were located in positions of the SARS-CoV-2 proteome with a high level of SNVs.

**Generation of ultraviolet (UV)-cleavable pHLA monomers.** The UV-cleavable peptides were synthesized in house by the Chemical Biology group, Leiden University Medical Centre. Recombinant HLA-A*01:01, A*02:01, A*03:01, A*11:01, A*24:02, B*07:02, B*08:01, B*15:01, B*18:01, and B*51:01 heavy chains and human beta-2 microglobulin (B2M) were produced in *Escherichia coli* and isolated from resulting inclusion bodies[58]. HLA-A and B heavy chains were refolded in the presence of B2M and UV-cleavable peptides (Supplementary Table 4) followed by gel filtration, biotinylation, and final purification by high-performance liquid chromatography[59].

**Generation of fluorescent pHLA multimers.** MHC complexes were loaded with the selected SARS-CoV-2 peptides via UV-induced ligand exchange[60,61]. In brief, pHLA complexes with UV-sensitive peptide were subjected to 254/366 nM UV light for 1 h at 4 °C in the presence of a rescue peptide. Fourteen fluorochrome–streptavidin reagents (Supplementary Table 5) were conjugated to pHLA monomers (100 μg/mL). For each pHLA monomer, conjugation was performed with 2 fluorochrome–streptavidin reagents resulting in dual fluorescent color codes for up to 75 epitopes. Subsequently, milk (Sigma, LP0031, 1% w/v) was added to block and capture unspecific peptide-binding residues, and fluorescently labeled pHLA multimers were incubated for 30 min on ice. Finally, D-biotin (Sigma, B4501, 26.3 mM) and NaN₃ (0.02% w/v) in phosphate-buffered saline (PBS) was added to block residual binding sites.

**Flow cytometry assays.** For the pHLA multimer assay and the phenotypic characterization, PBMC samples were thawed and washed with RPMI 1640 (Life Technologies, 21875-034) supplemented with human serum (Sigma, H3667, 10% v/v), penicillin–streptomycin (Life Technologies, 15140-122, 1% v/v), and benzonase nuclease (Merck-Millipore, 70746-4, 2500 U/mL); resuspended; and incubated at 37 °C for 30 min before staining. Antigen-specific CD8 T cells were stained for 15 min at 37 °C with pHLA multimers (Supplementary Table 6) encoding unique dual fluorescent code combinations for up to 75 epitopes. Subsequently, cells were stained for 20 min on ice with antibodies (Supplementary Table 6). LIVE/DEAD Fixable IR Dead Cell Stain Kit (Invitrogen, L10199) staining was performed for 20 min on ice either during antibody staining (pHLA multimer assay, 1/200) or for 10 min on ice after antibody staining (phenotypic characterization, 1/400). Individual staining was performed in the presence of Brilliant Staining Buffer Plus (BD, 563794) according to the manufacturer's protocol (BD), and samples were washed twice before acquisition.

**Analysis of SARS-CoV-2-specific CD8 T cell responses.** The following gating strategy shown in Supplementary Fig. 1 was applied to identify CD8⁺ T cells: (i) selection of live (IRDye low-dim) single-cell lymphocytes [forward scatter (FSC)-W/H low, side scatter (SSC)-W/H low, FSC/SSC-A], (ii) selection of anti-CD8⁺ and "dump" (anti-CD4, anti-CD14, anti-CD16, anti-CD19) negative cells. Antigen-specific CD8 T cell responses that were positive for only two and none of the other fluorescent pHLA multimers were identified using Boolean gating. Full gating strategy is shown. A minimum of 1000 CD8 T cells was required for further analysis. Cut-off values for the definition of positive antigen-specific CD8 T cell responses were ≥5 events and ≥0.005% of total CD8 T cells. To avoid experimental bias, analysis was carried out without prior knowledge about clinical patient characteristics, and to reduce researcher bias caused by manual gating, only positive responses that were confirmed by three independent researchers were defined as real. Data were analyzed using either the BD FACSDiva (v.8.0.1) or the FlowJo (v.10.6.2/10.7) software. To monitor the reproducibility of the assay system, reference samples with up to 10 CD8 T cell responses present at varying frequencies were included in each analysis.

**Peptide stimulation assay.** PBMCs were thawed, washed, and incubated at 37 °C for 30 min or at 4 °C for 60 min (if cells were simultaneously used for single-cell RNA sequencing) in RPMI 1640 (Life Technologies, 21875-034) supplemented with human serum (Sigma, H3667, 10% v/v), penicillin–streptomycin (Life Technologies, 15140-122, 1% v/v), and benzonase nuclease (Merck-Millipore, 2500 U/mL). After washing, equal amounts of PBMCs (≥1 × 10⁵ cells per condition) were cultured for 12 h at 37 °C in the presence of GolgiPlug (BD, 555029, 1/1000) and either the TTDPDFLGRY peptide (2 μg/mL) or equimolar amounts of DMSO (negative control). Phorbol 12-myristate 13-acetate (50 ng/mL) and Ionomycin (1 μg/mL) were used as technical control. Cells were washed and stained for 20 min on ice with surface marker antibodies (Supplementary Table 6). After washing, cells were stained for 10 min on ice with the LIVE/DEAD Fixable IR Dead Cell Stain Kit (Invitrogen, L10119, 1/400). Subsequently, cells were washed, fixed, and permeabilized using the Foxp3 Transcription Factor Staining Buffer Set (eBioscience, 00-5523-00) according to the manufacturer's protocol. Intracellular cytokines were stained for 20 min on ice with antibodies (Supplementary Table 6). Cells were washed twice before acquisition. The following gating strategy shown in Supplementary Fig. 3a–c was applied to identify CD8⁺ T cells using FlowJo (v.10.6.2/10.7): (i) selection of live (IRDye low-dim) single-cell lymphocytes [FSC-W/H low, SSC-W/H low, FSC/SSC-A], (ii) selection of anti-CD8⁺ and "dump" (anti-CD4, anti-CD14, anti-CD16, anti-CD19) negative cells. For the final analysis, acquired CD8 T cell counts of the DMSO control were normalized to the SARS-CoV-2 peptide condition. To quantify the frequency of TNF⁺, IFNγ⁺, IL-2⁺, and IL-17⁺ CD8⁺ cells, gates were set based on the DMSO control. Cut-off values for the definition of cytokine-producing CD8 T cell responses stimulated with the cognate SARS-CoV-2 peptide were ≥5 events and a ≥2-fold difference in the magnitude of TNF⁺, IFNγ⁺, IL-2⁺, or IL-17⁺ CD8⁺ cells compared to the DMSO control.

**Flow cytometer settings.** All samples were analyzed on the BD FACSymphony A5. The following 21-color instrument settings were used on the BD FACSymphony A5: blue laser (488 nm at 200 mW): FITC, 530/30BP, 505LP; BB630, 600LP, 610/20BP; BB700, 710/50BP, LP685; BB790, 750LP, 780/60BP. Red laser (637 nm at 140 mW): APC, 670/30BP, APC-R700, 690LP, 630/45BP, IRDye and APC-H7, 750LP, 780/60BP. Violet laser (405 nm at 100 mW): BV421, 420LP, 431/

28BP; BV480, 455LP, 470/20BP; BV605, 565LP, 605/40BP; BV650, 635LP, 661/11BP; BV711, 711/85, 685; BV750, 735LP, 750/30BP, BV786, 780/60BP, 750LP. UV laser (355 nm at 75 mW): BUV395, 379/28BP, BUV496, 515/30BP, 450LP; BUV563, 550LP, 580/20BP; BUV615, 600LP, 615/20BP; BUV661, 630LP, 670/25BP; BUV737, 735/44BP; 770LP; BUV805, 770LP, 819/44BP. Yellow-green laser (561 nm at 150 mW): PE, 586/15BP; PE Dazzle-594, 600LP, 610/20BP; PerCP-eF710, 710/50BP, 685LP; PE-Cy7, 750LP, 780/60BP. Appropriate compensation controls were included in each analysis.

**Statistical analysis.** Differences in the magnitude of identified CD8 T cell responses (stratified based on antigen source or recognized epitope) or the expression of inhibitory receptors between $CD8^+$ and $pHLA^+$ cells were assessed using non-parametric Mann–Whitney $U$ test. Phenotypic changes of SARS-CoV-2-specific CD8 T cells during acute COVID-19 disease and convalescence were assessed using Wilcoxon matched-pairs signed-rank test. Differences were considered significant if $P < 0.05$. Only significant $P$ values are displayed. Data cut-off for all analyses was 4 January 2021. Statistical analysis was performed using Excel (v.16.36) and PRISM 8 (v.8.4.0).

**Single-cell RNA and TCR sequencing.** PBMCs were thawed, washed, and incubated for 60 min on ice in cold RPMI 1640 (Life Technologies, 21875-034) supplemented with human serum (Sigma, H3667, 10% v/v), penicillin–streptomycin (Life Technologies, 15140-122, 1% v/v), and benzonase nuclease (Merck-Millipore, 70746-4, 2500 U/mL). TotalSeq$^{TM}$-streptavidin oligo barcoded reagents (Supplementary Table 5) were conjugated to pHLA monomers (100 μg/mL) as described above and used to stain SARS-CoV-2-specific CD8 T cells for 30 min on ice (Supplementary Table 6). Subsequently cells were stained for 20 min on ice with antibodies (Supplementary Table 6) and LIVE/DEAD Fixable IR Dead Cell Stain Kit (Invitrogen, L10119, 1/200). Stained cells from individual patients were pooled and washed before sorting on the FACSAria Fusion. The following gating strategy shown in Supplementary Fig. 4a was applied to identify and sort $CD8^+$ cells into PBS supplemented with bovine serum albumin (0.04% w/v) at 4 °C: (i) selection of live (IRDye low-dim) single-cell lymphocytes [FSC-W/H low, SSC-W/H low, FSC/SSC-A], (ii) selection of anti-$CD8^+$ positive and "dump" (anti-CD4, anti-CD14, anti-CD16, anti-CD19) negative cells.

The single-cell suspension was split into two samples that were successively loaded onto a Chromium Single Cell Chip (10× Genomics) according to the manufacturer's protocol for co-encapsulation with barcoded Gel Beads at a capture rate of ~1000 individual cells per sample. The following 10× Genomics kits were used to produce the Gel Bead-In Emulsions (GEMs) and the resulting sequence libraries (Gene expression library, Feature Barcode library, TCR library) according to the manufacturer's protocol: For batch I, Chromium Next GEM Single Cell 5' Library and Gel Bead Kit v1.1 (10× Genomics, PN-1000167), Chromium Next Gem Single Cell V(D)J Reagent Kit v1.1 (10× Genomics, PN-1000165), Chromium Single Cell 5' Library Construction Kit (10× Genomics, PN-1000020), Chromium Single Cell 5' Feature Barcode Library Kit (10× Genomics, PN1000080), and Chromium Next GEM Chip G Single Cell Kit (10× Genomics, PN-100127). For batch II, Chromium Next GEM Single Cell 5' Kit v2 (10× Genomics, PN-1000265), Chromium Next GEM Single Cell 5' Reagent Kits v2 (Dual Index) (10× Genomics, PN-2000263), Library Construction Kit (10× Genomics, PN-1000190), 5' Feature Barcode Kit (10× Genomics, PN-1000256), and Chromium Next GEM Chip K Single Cell Kit (10× Genomics, PN-1000287). The 3 libraries were combined in relative fractions of 0.785, 0.085, and 0.130 in order to generate sufficient reads per cell for each type of library. The final library pool was sequenced on a NextSeq Mid Flowcell, with 150 cycle chemistry kit in paired-end fashion 26-8-130 bp (batch I) or 26-10-10-130 bp (batch II). Full-length TCR V(D)J segments were enriched from amplified cDNA from 5' libraries via PCR amplification using: for batch I, the Chromium Single-Cell V(D)J Enrichment Kit (10× Genomics, PN-1000005), or for batch II, the Chromium Single Cell Human TCR Amplification Kit (10× Genomics, PN-1000252) according to the manufacturer's protocol (10× Genomics).

**Single-cell RNA and TCR sequencing data analysis.** The Cell Ranger Software Suite (v.3.1.0) was used to perform sample de-multiplexing, barcode processing, and single-cell 5' unique molecular identifier (UMI) counting. Single-cell RNA-sequencing data analysis was performed with Scanpy (v.1.5.1)[62]. Data were analyzed with Python (v.3.7.6)[63], pandas (v.1.0.1)[64] and NumPy (v.1.18.1)[65] were used for data manipulation, and Seaborn (v.0.10.0)[66] and Matplotlib (v.3.1.3)[67] were used for plotting. The single-cell transcriptome and TCR sequencing data were analyzed separately for batch I ($n = 5$, COVID-087, -096, -117, -143, and -153) and batch II ($n = 1$, COVID-131) to avoid the batch effect introduced by the use of the different Chromium Single Cell Chip. The following criteria were applied to each cell in batch I and batch II: gene count between 200 and 2500, mitochondrial gene percentage <0.25, and ribosomal gene percentage >0.2. Data were then normalized to depth 10,000, and $\ln(1 + x)$ was calculated. After filtering and normalization, the number of counts per cell and percentage of mitochondrial genes were regressed out from the data using scanpy.pp.regress_out. Data were subsequently scaled with scanpy.pp.scale using default parameters. Principal component analysis was computed on highly variable genes. A neighborhood graph of observations was computed with 50 principal components and n_neighbors = 10. UMAP plots were plotted using scaled data. Louvain clustering was performed with scanpy.tl.louvain with default parameters. Marker genes were found using scanpy.tl.rank_genes_groups on the non-scaled data (use_raw = True) with $t$ test. Differentially expressed genes were filtered based on a minimum ln fold change of >1 or <−1 and Benjamini–Hochberg false discovery rate value of <0.05. Differentially expressed genes that were upregulated or downregulated were used for gene ontology analysis using DAVID Bioinformatics Resources (v.6.8) (https://david.ncifcrf.gov/home.jsp) using default settings. TCR sequences for each single T cell were assembled by Cell Ranger vdj pipeline (v.3.1.0), leading to the identification of CDR3 sequences and the re-arranged TCR gene. TCR repertoire analysis was performed with Scirpy (v.0.3)[68]. TCR diversity and TCR clonal size were estimated by scirpy.tl.alpha_diversity and ir.pl.clonal_expansion (performing the normalization), respectively. V(D)J gene usage was estimated with scirpy.pl.vdj_usage. Abundance of particular TRB V segments was estimated with scirpy.pl.group_abundance, performing the normalization.

**Reporting summary.** Further information on research design is available in the Nature Research Reporting Summary linked to this article.

## Data availability
Single-cell RNA-sequencing data generated in this study are deposited in the Gene Expression Omnibus (GEO) repository under the accession code GSE169503. Flow cytometry data generated in this study are deposited in FlowRepository with the identifier FR-FCM-Z3KA. Source data are provided with this paper.

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

## Acknowledgements

These authors contributed equally: Andrea Cossarizza, Pia Kvistborg. We would like to thank Kees Korbee for his help with setting up a work environment according to the safety regulations; Ron Kerkhoven, Marja Nieuwland, Iris de Rink, Sacha Schepers, and Jessica Sieljes for the 10× Genomics data support; and Frank van Diepen, Anita Pfauth, and Martijn van Baalen for flow cytometry support with the BD FAC-Symphony, which was partly funded by the Louise Vehmeijer Stichting. We are grateful for the help from Stephanie Timmer and Patty Lagerweij for getting the COVID-19 patient samples shipped to Amsterdam. This work was partially supported by the Ministero della Salute, Bando Ricerca COVID-19 (grant number: COVID-2020-12371808, period: 2020-2021). S.D.B. and L.G. are Marylou Ingram scholars of the International Society for Advancement of Cytometry (ISAC) for the period 2015–2020 and 2020–2025, respectively.

## Author contributions

P.K. conceptualized and supervised this project. A.G., S.K., S.P., A.D., K.H., and P.K. carried out the experiments. A.G., S.K., and P.K. performed the flow cytometric data analysis. O.I. performed the single-cell RNA- and TCR-seq data analysis. A.G., S.K., O.I., S.D.B., A.C., and P.K. interpreted the data. S.D.B., L.G., C.M., G.G., M.G., and A.C. provided the patient samples and clinical data. C.M.P.T.O., P.J.M.H., M.T., R.B., T.N.M., and H.O. provided the reagents. A.G., S.K., O.I., and P.K. wrote the manuscript.

## Competing interests

The authors declare no competing interests.
