## [Peer Review File · Nature Communications]

REVIEWER COMMENTS

Reviewer #1 (Remarks to the Author):

The manuscript details the identification of SARS-CoV-2 derived CD8+ T cell epitopes using an array of predicted T cell epitopes and donors restricted by 10 common HLA I alleles. They observe dominant responses to epitopes within ORF1ab in many subjects, however these cells have a functional and transcriptional signature consistent with tight regulation / limited activation in vivo. The study is well executed and the findings solidly supported by the data, however the broader implications including clinical course of infection for different HLA allotypes and protective potential of these immunodominant CD8+ T cell responses await larger/future studies for confirmation. I have a few comments/questions:

1 – The subjects recruited in this cohort have quite severe COVID-19 infections. Do the authors have any comparable analysis of mild-moderate infections? Do the same immunodominance patterns apply? If this data is not available, the authors could expand the discussion to take into account that most individuals infected with SARS-CoV-2 have mild courses of infection with implications for T cell priming versus severe infection.

2 – The immunodominance of the HLA01*01-restricted TTD epitope was striking. The authors speculate that subjects with or lacking such alleles might be differentially susceptible to COVID-19. However, given this study relies upon predicted T cell epitopes, instead of an unbiased screen of the SARS-CoV-2 peptidome, it is possible that the algorithms might be more predictive for 01*01 than the other less-well characterized HLA studied, which could bias the peptide selection and confound the assignment of immunodominance hierarchies. Have the authors tested an overlapping peptide set (for some viral proteins at least) to validate their predictive algorithms are not missing, particularly for less common HLA alleles, potential potent CD8+ T cell responses in these donors?

3 -The authors demonstrate that poor cytokine production and/or activation is a feature of CD8 responses to COVID-19 and suggest this might limit contributions from CD8+ T cells to immunopathology. The authors should more fully discuss these observations in the context of publications from other groups characterizing mild-moderate infection

4 – The 10x scRNAseq data is interesting but some key details were unclear. While 5 patients were subject to sequencing but only 4 TTD responses analysed. This needs to be properly explained. What was the distribution of the TTD-specific CD8s between the subjects? The UMAP presentation for each individual would be informative to include as supplemental material.

5 – How confident can the authors be that the TTD-specific CD+ T cells analysed were actually recruited into the SARS-CoV-2 response given that only 180 differentially expressed genes were identified relative to naïve cells, versus some ~1500 differentially expressed genes relative to activated bulk CD8+ T cells.

Reviewer #2 (Remarks to the Author):

This manuscript describes an analysis of the SARS-Cov2 CD8 T cell s performed with a broad library of SARS-COV2 peptide/ HLA multimers in 18 patients with COVID-19.

The authors showed that this technology used to produce a large library of SARS-COV2 peptide/ HLA multimers (n=500) can be used to study SARs-COV2 specific CD8 T cell response

The authors claim that by using this method they were able to define an immunodominant response towards an ORF1 CD8 epitope and that the functionality of the CD8 T cells in COVID-19 patients is altered.

The work has some novelty since, characterization of the hierarchy and function of SARS-CoV2

specific CD8 T cells is still vastly incomplete. As such the analysis with a very broad quantity of HLA multimers has its value.

On the other hand, many conclusions are not supported by sufficient quantity of data.

The claim about immunodominance derives from data obtained in only 5 HLA-A01+ patients and the functional characterization of CD8 T cells is really weak.

The work is also generally difficult to read . Experiments are often poorly described and results are not displayed clearly .

Major Points:

A) The analysis of SARS-CoV2 CD8 T cells is done cross-sectionally, apparently at a single time point in patients with different severity of disease and at different time after onset of symptoms. The immunodominance of the HLA-A01:01 restricted epitope is tested only in 5 patients with only 2 healthy controls. To really demonstrate that response to this TTD epitope is immunodominant in HLA-A01+ patients the work should be expanded to test a large population of HLA-A01 patients and also healthy controls. In addition some of the subjects should be studied longitudinally to see whether the response persists after recovery.

B) The authors claim that the CD8 response to ORF1 is stronger than response to structural proteins. However the fact that they found more CD8 T cells specific for ORF epitope might be the reflection of the fact that 2/3 of their tetramers are done with ORF-1 based epitopes.

C) The authors analyze the TCR diversity of the TTD epitope response (figure 2d and e) to support the "T cell immunodominance" of such specific response. It is not clear what is their conclusion since they first claim TCR heterogeneity based on the sequence analysis of 32 CD8 T cells (mainly derived from single patient). However then they are saying that analysis of TCR beta usage shows an enrichment of TCR beta V27., which could mean that there is oligoclonality....The results appear at best inconclusive.

D) The claim related to the functional inability of CD8 T cells can be very important and somehow novel but it is not supported by any solid data. First the control experiments performed with T cells activated with PMA is not correct. T cell activation mediated with peptide is a TCR-mediated activation while PMA activation is not. As such the results about the lack of response to peptide stimulation lack any functional control. Authors should perform control experiments using unrelated peptides (ie. covering epitope of HCMV or EBV) or CD3 beads. It is also questionable why the authors choose to do a functional activation assay testing DR upregulation at 12 hours after stimulation. I think it is more appropriate to test cytokines production or activation markers upregulation at 4-6 hours. Furthermore, the data of the lack of cytokines production are not shown and the data related of DR expression show at best only two patients where a discrepancy is present. It is also not clear why the so called peptide pools are different for each subject. In some subject the peptide pools contain the single TTD peptide in other subjects no. This reviewer does not understand the logic of such experimental design. To demonstrate that CD8 T cells are dysfunctional during COVID-19 , more controlled experiments are necessary. CD8 T cells should also be tested at multiple time points to understand whether there is dynamic recovery of the function.

E) The single cell transcriptomic analysis describes mainly the results obtained from cells of a single patient (COVID-096)Figure 4, thus it can hardly be defined an unbiased characterization of TTD CD8 T cells of 5 patients.

We would like to thank the Editor and the Reviewers for their thoughtful comments. We have carefully addressed all comments and expanded the initial data set as described in detail below: 1) Thirteen additional COVID-19 patients were analyzed for CD8 T cell recognition of the selected SARS-CoV-2 epitopes. The updated data set now includes samples from 31 COVID-19 patients: critical (n=14), severe (n=10), moderate (n=2) and asymptomatic (n=5) disease (Table 1 and Table S2). The disease status has been updated according to the new WHO guidelines¹ for all patients (Table R1). In these patients, a total of 35 SARS-CoV-2-specific CD8 T cell responses specific for 18 different SARS-CoV-2 epitopes were detected (Fig. 1c). The results of the expanded data set confirm the immunodominant properties of the TTD epitope as described in more detail below and in the revised manuscript (page 4-5, line 123-145).

2) In terms of our functional data, we have (a) focused the functional analysis on TTD-specific CD8 T cell responses; (b) validated the lack of functionality for 2 of previously analyzed patients (Fig. R2); (c) obtained data on T cell functionality of 1 additional patient with acute COVID-19 disease (Fig. 3a and b); and (d) compared functionality during acute disease and convalescence for 2 patients (Fig. 5g and h). Together, these data confirm our previous findings demonstrating that TTD-specific CD8 T cells are dysfunctional during acute disease, and we now also demonstrate that TTD-specific CD8 T cells gain their functional capacity during convalescence as described in more detail below and in the revised manuscript (page 7-8, line 219-226).

3) We performed longitudinal and phenotypic analysis of identified SARS-CoV-2-specific CD8 T cell responses in 3 patients during acute disease and convalescence (Fig. 5). The results are described in detail in the revised manuscript (page 7-8, line 206-226).

Together, these data have strengthened the main conclusions of our work.

In addition, we have revised the introduction and discussion according to the current state-of-the-art literature and updated the analysis of mutational hotspots for the CD8 T cell recognized SARS-CoV-2 epitopes that have been identified in this study (page 4, line 106-121). All changes to the manuscript are highlighted.

Reviewer #1 (Remarks to the Author):

The manuscript details the identification of SARS-CoV-2 derived CD8⁺ T cell epitopes using an array of predicted T cell epitopes and donors restricted by 10 common HLA I alleles. They observe dominant responses to epitopes within ORF1ab in many subjects, however, these cells have a functional and transcriptional signature consistent with tight regulation/limited activation *in vivo*. The study is well executed, and the findings solidly supported by the data, however, the broader implications including clinical course of infection for different HLA allotypes and protective potential of these immunodominant CD8⁺ T cell responses await larger/future studies for confirmation. I have a few comments/questions:

We appreciate the reviewer comments highlighting the key findings of our work.

1. The subjects recruited in this cohort have quite severe COVID-19 infections. Do the authors have any comparable analysis of mild-moderate infections? Do the same immunodominance patterns apply? If this data is not available, the authors could expand the discussion to take into

account that most individuals infected with SARS-CoV-2 have mild courses of infection with implications for T cell priming versus severe infection.

We agree with the reviewer that it is intriguing to investigate the immunodominance pattern in patients with mild-moderate disease. Unfortunately, patients with mild disease recover in quarantine at home, and it was therefore not feasible to obtain peripheral blood samples from these patients. Similarly, patients with moderate disease are monitored closely but only rarely hospitalized². In our study, 2 patients with moderate disease were analyzed, however, these patients were not HLA-A*01:01⁺. Related studies performed their analyses with either samples collected during convalescence³⁻⁵ (our samples were analyzed during acute disease) and/or with the use of overlapping peptides^{3,4,6-8}. Due to the differences in the time of sample collection (convalescence vs acute disease) and the lack of information regarding the magnitude of individual CD8 T cell responses, it was therefore not possible to draw any conclusions about immunodominance patterns in COVID-19 patients with mild or moderate disease based on current literature.

It is important to note however, that the immunodominant properties of the TTD epitope were further strengthened in the expanded patient cohort in our study. In total, 9 HLA-A*01:01⁺ patients were analyzed (5 patients in the initial data set). The TTD-specific CD8 T cell response was detected in all HLA-A*01:01⁺ patients including patients with critical (n=4), severe (n=4), and asymptomatic (n=1) disease (Fig. 1b). To convey a clear message in the manuscript, we have focused our analysis on the immunodominance hierarchy in samples collected during acute disease (data from the asymptomatic patient was excluded for this purpose because the samples were collected during convalescence). First, our analysis confirmed that the magnitude of TTD-specific CD8 T cell responses was significantly higher compared to all other identified SARS-CoV-2-specific CD8 T cell responses across all patients (Fig. 2a, P=0.0003). Second, in 5 of the 8 HLA-A*01:01⁺ patients with acute disease additional SARS-CoV-2-specific CD8 T cell responses were detected (Fig. 2c), and the magnitude difference between the second highest magnitude response and the TTD-specific T cell response was on average >70 fold lower (Fig. 2c, P=0.0136). Third, TCR data analyses of TTD-specific CD8 T cells from 1 additional patient confirmed the high level of TCR diversity, which was previously observed in TTD-specific CD8 T cells of 4 patients (Fig. 2d).

2. The immunodominance of the HLA01*01-restricted TTD epitope was striking. The authors speculate that subjects with or lacking such alleles might be differentially susceptible to COVID-19. However, given this study relies upon predicted T cell epitopes, instead of an unbiased screen of the SARS-CoV-2 peptidome, it is possible that the algorithms might be more predictive for 01*01 than the other less-well characterized HLA studied, which could bias the peptide selection and confound the assignment of immunodominance hierarchies. Have the authors tested an overlapping peptide set (for some viral proteins at least) to validate their predictive algorithms are not missing, particularly for less common HLA alleles, potential potent CD8+ T cell responses in these donors?

We understand the reviewers concern regarding potential prediction bias for HLA-A*01:01. We have considered an experimental unbiased approach to validate our predicted epitopes by generating all potential overlapping 9-11mers from a part of the viral proteome and loading these epitopes on antigen presenting cells for subsequent *ex vivo* stimulation of CD8 T cells.

However, 2 major practical limitations made this approach unachievable. First, such analysis would require a substantial amount of CD8 T cells. Due to both ethical concerns and the lymphopenia frequently present in COVID-19 patients, it was unfortunately not feasible to obtain sufficient cell numbers from these patients. Second, given the lack of functionality of SARS-CoV-2 specific CD8 T cell responses (Fig. 3a and b), responses may be missed, in particular those not related to common cold coronaviruses.

We therefore performed an *in-silico* bioinformatics approach to evaluate the coverage of immunogenic epitopes using our prediction strategy for Influenza A virus (FLU) and Epstein-Barr virus (EBV). A total of 50 epitopes were selected for each of the HLA alleles included in our study, which overlapped with information regarding confirmed T cell epitopes for EBV and FLU, respectively. Based on the Immune Epitope Data Base (IEDB), 23 (restricted by 7/10 included HLA alleles) and 31 (restricted by 6/10 included HLA alleles) immunogenic epitopes were reported for FLU and EBV, respectively. On average, 76% (range: 50% to 100% for individual alleles) of the immunogenic epitopes derived from FLU were covered by our predictions across 7/7 HLA alleles for which immunogenic epitopes have been reported (Fig. R1a). For EBV, on average 38% (range: 0% to 100% for individual alleles) of the immunogenic epitopes were covered by our predictions across 4/6 HLA alleles (Fig. R1b). In addition, 4/4 and 3/6 of the immunodominant epitopes were covered by our predictions for FLU and EBV, respectively (Fig. R1c). The coverage of immunogenic and immunodominant epitopes was lower for EBV in comparison to FLU, which is likely due to the significant difference in size of the 2 viral genomes (FLU: 13.5 kb, EBV: 170 kb) with FLU being more similar in size to SARS-CoV-2 (30kb). Overall, these analyses show a broad coverage of immunogenic and immunodominant epitopes across different alleles, even though the coverage is imperfect. It is unlikely that the overrepresentation of CD8 T cell recognized epitopes restricted by HLA-A*01:01 compared to CD8 T cell recognized epitopes restricted by other HLA alleles is only based on differences in prediction quality across HLA alleles. We agree with the reviewer, however, that we cannot formally make the claim that our predictions were not biased (page 5, line 131-133 in initial manuscript) and have therefore corrected this in the revised manuscript. To provide additional transparency regarding our prediction strategy, we have clarified the prediction pipeline in the manuscript (page 11-12, line 348-361) and provided an overview of the prediction scores for proteasomal processing (NetChop-3.1 scores) and for predicted binding affinity (NetMHCpan-4.0 scores) for the 50 selected epitopes and the 18 CD8 T cell recognized epitopes for each HLA allele (Fig. S6).

3. The authors demonstrate that poor cytokine production and/or activation is a feature of CD8 responses to COVID-19 and suggest this might limit contributions from CD8⁺ T cells to immunopathology. The authors should more fully discuss these observations in the context of publications from other groups characterizing mild-moderate infection.

We thank the reviewer for this comment. We have put our data in the context of other studies that have analyzed samples from patients mild-moderate disease, which is described in detail the discussion (page 9, line 281-289). However, it is important to note that data on the functionality of SARS-CoV-2-specific CD8 T cells of COVID-19 patients with acute mild-moderate disease is currently lacking. To further strengthen our own data, we have validated the functionality results for 2 of the previously included patients (COVID-143 and COVID-

153, Fig. R2). Furthermore, we were able to obtain sufficient numbers of CD8 T cells (criteria described in detail in the revised manuscript on page 15, line 454-459) for 1 additional patient with critical disease. For clarity, we have focused the functionality data on TTD-specific CD8 T cell responses resulting in data from a total of 5 patients. The majority of analyzed patients with acute disease (4/5) lacked the capacity to produce cytokines upon stimulation with the TTD epitope (Fig. 3a and b). To convey a clear message in the manuscript, we do not include the data of the asymptomatic patient because the samples were collected 3 months after the positive PCR test for SARS-CoV-2. It is, however, noteworthy that the SARS-CoV-2-specific CD8 T cells from the asymptomatic patient were functional (Fig. R3). These data are in line with our functional data on TTD-specific CD8 T cells during acute COVID-19 disease and convalescence of 2 patients (COVID-117 and COVID-143), which show that TTD-specific CD8 T cells gain functionality during convalescence (Fig. 5g and h).

4. The 10x scRNAseq data is interesting but some key details were unclear. While 5 patients were subject to sequencing but only 4 TTD responses analyzed. This needs to be properly explained. What was the distribution of the TTD-specific CD8s between the subjects? The UMAP presentation for each individual would be informative to include as supplemental material.

We apologize for the ambiguity regarding the patient numbers. The reason that TTD responses were analyzed from 4 patients out of 5 is because 1 of the patients (COVID-087) was HLA-A*01:01 negative and therefore lacked the TTD-specific CD8 T cell response. We have clarified this in the manuscript (page 6, line 175-178) and provided an overview of the cell numbers (total CD8 T cells and TTD-specific CD8 T cells) for each patient (Table 2). Because the cell numbers for some of the patients are low, individual UMAPs are not useful. However, we have included a UMAP showing the distribution of cells for each patient in Fig. S4a. Please note, that data of 1 additional patient has been included in the revised manuscript (page 6-7, line 168-204).

5. How confident can the authors be that the TTD-specific CD8⁺ T cells analyzed were actually recruited into the SARS-CoV-2 response given that only 180 differentially expressed genes were identified relative to naïve cells, versus some ~1500 differentially expressed genes relative to activated bulk CD8⁺ T cells.

We thank the reviewer for this comment. A few key observations support the hypothesis that the TTD-specific CD8 T cells were recruited during acute COVID-19 infection. First, we observe a profound magnitude (on average 7% of total CD8 T cells, range: 0.074% to 19%) of the TTD-specific CD8 T cell response in patients with acute disease (Fig. 2a, and Table S3) suggesting that these cells have been actively expanding during the infection. Furthermore, the longitudinal data of SARS-CoV-2-specific CD8 T cell responses (n=6) from 3 patients (COVID-096, COVID-117, COVID-143) included in the revised manuscript (page 7-8, line 220-226) show a decrease in the magnitude of the TTD-specific CD8 T cell response during convalescence (average fold-change: 9-fold, range 1.2-2.2) (Fig. 5a and b). In line with the decrease in magnitude, phenotypic analysis revealed that TTD-specific CD8 T cells convert from effector-like cells to memory cells during convalescence. Specifically, we found a significant decrease in the percentage of activated cells based on expression of HLA-DR and

CD95 and an increase in the percentage of memory CD8 T cells expressing CD45RA and CCR7 but lower CD27 compared to naïve cells (Fig. 5c and d). Second, analysis of the single cell gene expression data of the initial (batch I) and expanded (batch II) datasets showed that TTD-specific CD8 T cells display a clear activation program compared to naïve CD8 T cells. Differential gene expression analysis resulted in 97 genes that were significantly upregulated and 33 genes that were significantly downregulated in TTD-specific CD8 T cells compared to naïve CD8 T cells in both data batches. Gene ontology analysis of differentially expressed genes that were upregulated in TTD-specific CD8 T cells compared to naïve CD8 T cells and overlapping between the 2 data sets revealed 23 significantly differentially regulated processes related to T cell activation (GO:0050776~regulation of immune response, GO:0006955~immune response, GO:0050852~T cell receptor signaling pathway, GO analysis for batch I and batch II are shown in Table S7).

Reviewer #2 (Remarks to the Author):

This manuscript describes an analysis of the SARS-Cov2 CD8 T cells performed with a broad library of SARS-COV2 peptide/ HLA multimers in 18 patients with COVID-19. The authors showed that the technology used to produce a large library of SARS-COV2 peptide/ HLA multimers (n=500) can be used to study SARs-COV2 specific CD8 T cell response. The authors claim that by using this method they were able to define an immunodominant response towards an ORF1 CD8 epitope and that the functionality of the CD8 T cells in COVID-19 patients is altered. The work has some novelty since, characterization of the hierarchy and function of SARS-CoV2 specific CD8 T cells is still vastly incomplete. As such the analysis with a very broad quantity of HLA multimers has its value. On the other hand, many conclusions are not supported by sufficient quantity of data. The claim about immunodominance derives from data obtained in only 5 HLA-A01+ patients and the functional characterization of CD8 T cells is really weak. The work is also generally difficult to read. Experiments are often poorly described, and results are not displayed clearly.

We thank the reviewer for appreciating the use of a broad approach to study the SARS-CoV-2 specific CD8 T cell responses as well as the novelty of our work. We have improved the description of the experiments and displayed the data more clearly.

1. The analysis of SARS-COV2 CD8 T cells is done cross-sectionally, apparently at a single time point in patients with different severity of disease and at different time after onset of symptoms. The immunodominance of the HLA-A01:01 restricted epitope is tested only in 5 patients with only 2 healthy controls. To really demonstrate that response to this TTD epitope is immunodominant in HLA-A01+ patients the work should be expanded to test a large population of HLA-A01 patients and also healthy controls. In addition, some of the subjects should be studied longitudinally to see whether the response persist after recovery.

We apologize for the lack of clarity regarding the time for the sample collection. All samples of patients with acute moderate, severe and critical disease were collected during hospitalization/acute infection. The median duration of the hospitalization before sample collection was 12 days ranging from 2 days before hospitalization and 24 days post hospitalization (Table 1 and Table S2). Samples from COVID-19 patients with asymptomatic

disease were collected during convalescence (3 month post positive PCR test for SARS-CoV-2) because these patients were at home in quarantine during the infection period.

We understand the concern regarding the limited size of our initial data set. We have therefore expanded the analysis of SARS-CoV-2-specific CD8 T cells to 31 COVID-19 patients in total including patients with critical (n=14), severe (n=10), moderate (n=2) and asymptomatic (n=5) disease and 3 additional healthy donors (total, n=7) (Fig. 1c). Of the 13 additional patients, 4 patients were HLA-A*01:01⁺ resulting in a final data set of 9 HLA-A*01:01⁺ patients. The TTD-specific CD8 T cell response was detected in all 9 HLA-A*01:01⁺ patients with critical (n=4), severe (n=4) and asymptomatic (n=1) disease (Fig. 1c). In addition, we analyzed 2 extra samples from HLA-A*01:01⁺ healthy donors resulting in 4 HLA-A*01:01⁺ healthy controls in total. In contrast to COVID-19 patients, no TTD-specific CD8 T cell responses were detected in HLA-A*01:01⁺ healthy donors (n=4). The sampling time point for the asymptomatic patient (during convalescence) may explain the low magnitude (0.011% of total CD8⁺ cells) of the TTD-specific CD8 T cell response detected in this patient whereas samples from patients with critical and severe disease were collected during acute infection (Table 1). To convey a clear message in the manuscript, we have focused our analysis of the immunodominance hierarchy on samples collected during acute disease. First, our analysis confirmed that the magnitude of TTD-specific CD8 T cell responses was significantly higher compared to all other identified SARS-CoV-2 specific CD8 T cell responses across patients (Fig. 2a, P=0.0003). Second, in 5 of the 8 HLA-A*01:01⁺ patients, we have identified additional SARS-CoV-2 specific CD8 T cell responses besides the TTD-specific CD8 T cell response (Fig. 2b), and the magnitude of these TTD-specific CD8 T cells responses was on average >70 fold higher compared to the second highest magnitude response per patient (Fig. 2c, P=0.0136). Together, these data confirm our previous findings, which support the immunodominant property of the TTD epitope.

To address the request for longitudinal analysis, we have analyzed samples from 3 COVID-19 patients during acute disease and convalescence (4 to 5 months post hospital discharge). These results show a decrease in the magnitude of the TTD-specific CD8 T cell responses during convalescence (average fold-change: 9, range 1.2 to 2.2) (Fig. 5a and b). In line with the decrease in magnitude, phenotypic analysis revealed that TTD-specific CD8 T cells convert from effector cells to memory cells during convalescence compared to acute disease. Specifically, we found a decrease in the percentage of activated cells based on expression of HLA-DR and CD95 and an increase in percentage of memory-like CD8 T cells expressing CD45RA and CCR7 (Fig. 5c and d). Furthermore, we show that SARS-CoV-2-specific CD8 T cells gain functionality during convalescence (Fig. 5g and h). The results are described in detail in the manuscript (page 7-8, line 206-226).

2. The authors claims that the CD8 response to ORF1 is stronger than response to structural proteins. However, the fact that they found more CD8 T cells specific for ORF epitope might be the reflection of the fact that 2/3 of their tetramers are done with ORF-1 based epitopes.

We thank the reviewer for making this relevant point. It is indeed correct that we have included a higher fraction of epitopes derived from the ORF1ab (74% of total selected epitopes, Fig. 1d) in comparison to other ORFs of SARS-CoV-2 (26% of selected epitopes, Fig. 1g). The high number of selected epitopes derived from the ORF1ab reflects the large size of the ORF1ab in

comparison to the other viral proteins, which are considerably smaller in size (Fig. 1g). However, the epitopes recognized by CD8 T cells do not reflect the size of individual ORFs. We observe an enrichment for CD8 T cell recognized epitopes derived from the spike protein (Fig. 1g). These responses are of overall lower magnitude compared to CD8 T cell responses specific for epitopes derived from the ORF1ab (Fig. 1e). The lack of high magnitude responses specific for (immunodominant) epitopes derived from the structural proteins, however, may indeed be related to our predictions and the selection of only 50 epitopes per HLA allele. We are aware that we have only identified 1 of potentially many immunodominant epitopes from SARS-CoV-2, and we have clarified this point in the discussion (page 8, line 241-244).

3. The authors analyze the TCR diversity of the TTD epitope response (figure 2d and e) to support the “T cell immunodominance” of such specific response. It is not clear what is their conclusion since they first claim TCR heterogeneity based on the sequence analysis of 32 CD8 T cells (mainly derived from single patient). However, then they are saying that analysis of TCR beta usage shows an enrichment of TCR beta V27, which could mean that there is oligoclonality.... The results appear at best inconclusive.

We thank the reviewer for this comment and apologize for the lack of clarity. Our conclusion on the high TCR diversity shown in Fig. 2d was based on the sequencing data of the TRB-CDR3, which is considered to be the most important part for the interaction with the cognate peptide major histocompatibility complex that defines the clonal diversity within antigen-specific CD8 T cell responses⁹. Despite the high TCR diversity of the TRB-CDR3 sequences, we observed an enrichment of TRBV27 within the TTD-specific CD8 T cell response in comparison to bulk CD8 T cells. The TRBV segment diversity is limited to ~50 segments and is therefore often shared between T cell clones⁹. Some TCR rearrangements occur more frequently resulting in specific segments being used more frequently¹⁰. There are a number of examples demonstrating a high CDR3 heterogeneity but biased TRBV segment usage for TCRs specific for immunodominant viral epitopes. For example, one study showed that 70% of ~10,000 unique TRB-CDR3 sequences isolated from CD8 T cells specific for the immunodominant FLU epitope GILGFVFTL restricted by HLA-A*02:01 share the TRBV19 region¹¹.

4. The claim related to the functional inability of CD8 T cells can be very important and somehow novel, but it is not supported by any solid data. First the control experiments performed with T cells activated with PMA is not correct. T cell activation mediated with peptide is a TCR-mediated activation while PMA activation is not. As such the results about the lack of response to peptide stimulation lack any functional control. Authors should perform control experiments using unrelated peptides (ie. covering epitope of HCMV or EBV) or CD3 beads. It is also questionable why the authors choose to do a functional activation assay testing DR upregulation at 12 hours after stimulation. I think it is more appropriate to test cytokines production or activation markers upregulation at 4-6 hours. Furthermore, the data of the lack of cytokines production are not shown and the data related of DR expression show at best only two patients were a discrepancy is present. It I also not clear why the so called peptide pools are different for each subjects. In some subject the peptide pools contain the single TTD peptide in other subjects no. This reviewer does not understand the logic of such experimental design.

To demonstrate that CD8 T cells are dysfunctional during COVID-19, more controlled experiments are necessary. CD8 T cells should also be tested at multiple time points to understand whether there is dynamic recovery of the function.

We thank the reviewer for these comments regarding the functional data of the SARS-CoV-2 specific CD8 T cell responses.

Stimulation with PMA/IO was used as a technical control of the assay to ensure that a potential lack of cytokine production upon peptide stimulation was not due to any technical aspects. This is now clarified in the revised manuscript (page 5, line 152-153). To address the comment about the use of CD3 stimulation and the duration of the stimulation, we have performed a comparison between the 2 conditions (PMA/IO and plate bound CD3) at 6h and 12h using material from 2 COVID-19 patients with severe/critical disease. Our results show that cells stimulated with plate bound CD3 lacked IL-2 production, while the signal for both IFN γ and TNF α were significantly lower compared to the PMA/IO control (Fig. R4). These data show that stimulation with PMA/IO is a suitable technical control even though it does not provide information regarding biology.

We agree with the reviewer that it would be useful to add viral peptides from other viruses. We therefore included a selection of epitopes derived from other viruses when we screened for SARS-CoV-2-specific CD8 T cell responses in 13 newly included patients. We detected 4 CMV-specific responses in 4 patients, an EBV-specific response in 1 patient and a FLU-specific CD8 T cell response in 1 patient. We have included these epitopes in the functional assay for 3 patients and were able to obtain sufficient numbers of CMV-specific CD8 T cells for 1 patient. Stimulation with the CMV-derived epitope restricted to HLA-A*01:01 showed that these cells were functional (Fig. R5a and b). However, we have not included these data in the manuscript because the results are limited to only 1 patient.

Regarding the use of peptide pools, we apologize for the lack of proper explanation. We had initially included samples stimulated with several peptides in order to ensure that we have sufficient numbers of SARS-CoV-2-specific CD8 T cells to detect reactivity upon peptide stimulation if present. For clarity, we have focused our analysis on cytokine production in samples stimulated with the TTD epitope alone (Fig. 3a and b). This has been clarified in the results (page 5, line 150) and materials and methods (page 14, line 437-440).

The functional data is now displayed in the main figure (Fig. 3a and b), including new data of 1 additional patient (COVID-131) with critical COVID-19 disease in which SARS-CoV-2 specific CD8 T cells were found to be functional (2.7% IFN γ and 0.3% TNF α) although at a much lower magnitude compared to the pHLA multimer stain (19% of total CD8⁺ cells). To validate the initial functional data, samples from patient COVID-143 and 153 were re-analyzed and the results confirmed the initial results (Fig. R2).

Finally, we have assessed the functionality of the TTD-specific CD8 T cell response at the time of acute infection as well as at the time of convalescence (4 to 5 months post hospital discharge) for 2 patients (COVID-117 and COVID-143), and these data revealed that the TTD-specific CD8 T cell responses gain functionality at the time of convalescence (Fig. 5g and h).

5. The single cell transcriptomic analysis describes mainly the results obtained from cells of a single patient (COVID-096) Figure 4; thus, it can hardly be defined an unbiased characterization of TTD CD8 T cells of 5 patients.

We thank the reviewer for this fair point. We have now clarified in the manuscript that unbiased characterization refers to a broad characterization, which is not based on a panel of pre-selected markers such as phenotypic analysis based on flow cytometry but rather the assessment of all genes found to be expressed based on single cell RNA sequencing (page 6, line 168-171).

As pointed out by the reviewer, the vast majority of TTD-specific CD8 T cells were from 1 COVID-19 patient. For clarification, we have included a table with the number of bulk CD8 T cells and TTD-specific T cells per patient in the revised manuscript (Table2). Please note that we have obtained single cell RNA sequencing data from 1 additional patient (COVID-131). The new data are described more in detail in the manuscript (Fig. 4, page 6-7, line 168-204).

References

1. (null), W. H. O. (. H. Clinical management of COVID-19: Interim guidance. 1–62 (2020).
2. Gandhi, R. T., Lynch, J. B. & del Rio, C. Mild or Moderate Covid-19. *N Engl J Med* **383**, 1757–1766 (2020).
3. Nelde, A. *et al.* SARS-CoV-2-derived peptides define heterologous and COVID-19-induced T cell recognition. *Nature Immunology* **8**, 1–25 (2020).
4. Sekine, T. *et al.* Robust T Cell Immunity in Convalescent Individuals with Asymptomatic or Mild COVID-19. *Cell* **183**, 158–168.e14 (2020).
5. Schulien, I. *et al.* Characterization of pre-existing and induced SARS-CoV-2-specific CD8+T cells. *Nat Med* **27**, 78–85 (2021).
6. Weiskopf, D. *et al.* Phenotype and kinetics of SARS-CoV-2-specific T cells in COVID-19 patients with acute respiratory distress syndrome. *Sci Immunol* **5**, eabd2071–11 (2020).
7. Braun, J. *et al.* SARS-CoV-2-reactive T cells in healthy donors and patients with COVID-19. *Nature* **587**, 270–274 (2020).
8. Sattler, A. *et al.* SARS-CoV-2-specific T cell responses and correlations with COVID-19 patient predisposition. *J. Clin. Invest.* **130**, 6477–6489 (2020).
9. Freeman, J. D., Warren, R. L., Webb, J. R., Nelson, B. H. & Holt, R. A. Profiling the T-cell receptor beta-chain repertoire by massively parallel sequencing. *Genome Research* **19**, 1817–1824 (2009).
10. Lanzarotti, E., Marcatili, P. & Nielsen, M. T-Cell Receptor Cognate Target Prediction Based on Paired α and β Chain Sequence and Structural CDR Loop Similarities. *Front. Immunol.* **10**, 2080–10 (2019).
11. Chen, G. *et al.* Sequence and Structural Analyses Reveal Distinct and Highly Diverse Human CD8+TCR Repertoires to Immunodominant Viral Antigens. *CellReports* **19**, 569–583 (2017).

Figures Rebuttal

Fig. R1: Predicted coverage of immunogenic and immunodominant epitopes derived from FLU and EBV. Employing the prediction strategy used for the SARS-CoV-2 epitope prediction and selection on the proteomes of FLU and EBV, a total of 50 epitopes were predicted for 7 of the 10 HLA alleles used in the SARS-CoV-2 analysis for FLU and 6 HLA alleles for EBV. The number of selected HLA alleles included for FLU and EBV was based on information available about immunogenic and immunodominant epitopes from each virus from IEDB.

- Coverage (shown as percentage on y-axis, actual numbers are shown on top for each HLA allele) of immunogenic FLU-derived epitopes restricted to the 7 HLA alleles which were included (dark blue) in our selection or not (light blue).
- Coverage (shown as percentage on y-axis, actual numbers are shown on top for each HLA allele) of immunogenic EBV-derived epitopes restricted to the 6 HLA alleles which were included (dark blue) in our selection or not (light blue).
- Coverage (shown as percentage on y-axis, actual numbers are shown on top for each HLA allele) of immunodominant FLU- and EBV-derived epitopes that were included (dark blue) or not included (light blue) within our selection.

Fig. R2: Validation of absent cytokine production in CD8 T cells of COVID-19 patients with acute disease across 2 independent experiments.

- a) Representative gating strategy used to assess the functional capacity of TTD-specific CD8 T cells in COVID-143 across 2 independent experiments. Cells were stimulated for 12h with DMSO (negative control) and either with a SARS-CoV-2 peptide pool (TTD, CTD, PTD, DTD, experiment 1) or the TTD peptide alone (experiment 2). The gates were set based on the background signal in the DMSO control.
- b) Lack of cytokine production upon stimulation with SARS-CoV-2 peptides across 2 independent experiments for 2 patients (COVID-143 and COVID-153) with acute severe and critical disease.

Fig. R3: Functionality of SARS-CoV-2-specific CD8 T cell responses from a convalescent COVID-19 patient with asymptomatic disease.

- Representative gating strategy used to assess the functional capacity of TTD-specific CD8 T cells in COVID-219. Cells were stimulated for 12h with DMSO (negative control) or the KTF peptide (KTFPPTEPK restricted to HLA*A-03:01). The gates were set based on the background signal in the DMSO control.
- SARS-CoV-2-specific CD8 T cells from an asymptomatic COVID-19 patient expressing cytokines after 12h stimulation with the KTF peptide. Percentages represent the frequency of cytokine producing cells after subtracting the percentages of the DMSO control. Samples were collected during convalescence (3 months post positive PCR test for SARS-CoV-2).

Fig. R4: Evaluation of control conditions using different stimuli (PMA/IO vs plate bound CD3) and time points (6h vs 12h).

PBMCs from 2 COVID-19 patients were cultured in triplicates without or with PMA/IO or plate bound CD3. Functionality was assessed after 6h and 12h based on IFN γ , TNF α , IL-2 and IL-17. Unstim.: unstimulated.

Fig. R5: Functionality of a CMV-specific CD8 T cell response from a COVID-19 patient with acute critical disease.

- Representative gating strategy used to assess the functional capacity of CMV-specific CD8 T cells reactive in COVID-131. Cells were stimulated for 12h with DMSO (negative control) or the VTE peptide (VTEHDTLLY restricted to HLA*A-01:01). The gates were set based on the background signal in the DMSO control.
- CMV-specific CD8 T cells of a COVID-19 patient with acute critical disease expressing cytokines after 12h stimulation with the VTE peptide. Percentages represent the frequency of cytokine producing cells after subtracting the percentages of the DMSO control.

Tables

Table R1. Initial and updated disease status according to WHO guidelines for all patients included in this study¹.

Data set	Patient	Disease status (initial)	Disease status (updated)
Initial	COVID-042	Critical	Critical
Initial	COVID-094	Critical	Critical
Initial	COVID-112	Critical	Critical
Initial	COVID-121	Severe (oxygen)	Critical
Initial	COVID-123	Critical	Critical
Initial	COVID-127	Severe (oxygen)	Critical
Initial	COVID-129	Critical	Critical
Initial	COVID-140	Critical	Critical
Initial	COVID-141	Critical	Critical
Initial	COVID-143	Severe (no oxygen)	Critical
Initial	COVID-147	Critical	Critical
Initial	COVID-150	Critical	Critical
Initial	COVID-152	Severe (oxygen)	Critical
Initial	HD-01	Healthy	Healthy
Initial	HD-02	Healthy	Healthy
Initial	HD-03	Healthy	Healthy
Initial	HD-05	Healthy	Healthy
Initial	COVID-002	Severe (oxygen)	Moderate

Initial	COVID-004	Severe (oxygen)	Moderate
Initial	COVID-009	Severe (oxygen)	Severe
Initial	COVID-033	Severe (oxygen)	Severe
Initial	COVID-087	Severe (oxygen)	Severe
Initial	COVID-096	Severe (no oxygen)	Severe
Initial	COVID-116	Severe (oxygen)	Severe
Initial	COVID-117	Severe (oxygen)	Severe
Initial	COVID-153	Severe (no oxygen)	Severe
Extended	COVID-218	N/A	Asymptomatic
Extended	COVID-219	N/A	Asymptomatic
Extended	COVID-220	N/A	Asymptomatic
Extended	COVID-221	N/A	Asymptomatic
Extended	COVID-222	N/A	Asymptomatic
Extended	COVID-223	N/A	Asymptomatic
Extended	COVID-224	N/A	Asymptomatic
Extended	COVID-024	N/A	Critical
Extended	COVID-040	N/A	Critical
Extended	COVID-084	N/A	Critical
Extended	COVID-131	N/A	Critical
Extended	COVID-180	N/A	Critical
Extended	HD-06	N/A	Healthy
Extended	HD-07	N/A	Healthy
Extended	HD-08	N/A	Healthy
Extended	HD-10	N/A	Healthy
Extended	COVID-007	N/A	Severe
Extended	COVID-015	N/A	Severe
Extended	COVID-111	N/A	Severe
Extended	COVID-166	N/A	Severe
Extended	COVID-174	N/A	Severe

REVIEWER COMMENTS

Reviewer #1 (Remarks to the Author):

The authors have addressed the majority of my concerns, with the impressive generation of new experimental data and analysis. I am happy to recommend publication. I do still have concern over the use of "immunodominant" in the title, as I do not think this has been comprehensively established by this study and is a heavily loaded term. I suggest modification.

Reviewer #2 (Remarks to the Author):

The authors partially answered my questions:

Point 1;

The immunodominance of the TTD-HLA-A01-restricted CD8 T cell epitope is a very interesting point of this paper and the fact that such CD8 T cell response was detected at frequency that appears to be around 10-20% of total CD8 T cells (3 out of 8 patients) and among 1-10% of total CD8 T cells is quite extraordinary (Figure 1C) . However I don't understand why that the authors do not somehow point out that such incredibly high frequency directly ex vivo were never found in SARS-CoV2 infected patients studied so far (it is only mentioned briefly in the discussion) . Classical frequency of CD8 T cells during acute disease is always less than 0.5%. This is why I think it will be important to show more control and to test in more HLA-A01+ SARS-CoV2 patients the behavior of such response. The authors partially answer to my question and they add additional 4 HLA-A01+ SARS-Cov2 patients and 2 HLA-A01 healthy controls. It is better than nothing but certainly does not address for example the problem of why such high frequencies of TTD-specific CD8 T cells were detected almost exclusively in critical and severe patients. In reality if we look at the data, the real interesting point is that the authors found extremely high frequency of TTV-HLA-A01-specific CD8 T cells only in severe/critical patients while frequency of this TTD-specific CD8 T cells is very low in the only HLA-A01 asymptomatic HLA-A01+ patient studied. I still think that a larger quantity of HLA-A01+ SARS-CoV-2 infected patient with mild and asymptomatic infection should be studied to better understand whether high frequencies of such TTV-CD8 T cells are present only in severe/critical cases. I also think that a complete set of dot plots with appropriate controls (ot should be shown to display such high frequency of TTV-specific CD8 T cells . Controls of the staining of TTV-A01+ CD8 T cells in non-HLA-A01+ severe patients should also be shown to see the level of background staining detected.

Some longitudinal data were also added and they are also very interesting (Figure 5) . Frequency (patient 143 and 117) of the TTV-A01-restricted CD8 T cells remains very high (in 117 remains at about 2% of total CD8 T cells after recovery) and in patient 143 remains superior to 1% of total Ccd8 T cells. It is not very clear why the authors are describing such data saying that " there is a decrease of the magnitude of such CD8 T cell response". This is certainly true in patients 096 but it is not true for 117 and the persistence of a CD8 T cell response specific for a single epitope at a frequency higher than 1% (patient 143) is certainly not so common.

This is why I still think that the CD8 T cell response against this epitope , that is the single focus of this work should be better analyzed in more subjects.

POINT 2.

The authors agree than their data cannot be used to generally conclude that ORF-1 is more immunogenic in the totality of the patients. This could happen only in HLA-A01+ subjects. They have modified the text. However in the abstract they are writing that "A total of 18 SARS-CoV-2 epitopes were identified; 8 of these epitopes were derived from the ORF1ab". The sentence should be changed since it seems to imply that ORF-1 is in general more immunogenic"

POINT 3. The experiments of functionality remains problematic. It is likely that the CD8 T cells analyzed during the acute phase of severe COVID-19 were functional impaired . These data are in line with recent results showing that "functional SARS-CoV-2 specific T cell response is present during acute infection only in subjects with very mild infection and not in severe cases. (Rydzynski Moderbacher C, et al. Antigen-Specific Adaptive Immunity to SARS-CoV-2 in Acute COVID-19 and Associations with Age and Disease Severity. Cell 2020; 183: 996–1012.e19. and Tan AT, et al.

Early induction of functional SARS-CoV-2-specific T cells associates with rapid viral clearance and mild disease in COVID-19 patients. Cell Reports 2021; 53: 108728–13.). These data should be quoted and the results discussed in the context of these new findings while the discussion about T cells are not functional because they are not cross-reactive with other common coronavirus (page 9 , line 276-86) is quite implausible(and are we sure that the TTV sequence is not shared among other pathogens?). In addition, the lack of control of such functional experiments remains. To demonstrate that SARS-COV-2-specific CD8 T cells are the only T cells that are not functional during severe COVID-19, the control experiments performed with peptides of other viruses (included Figure R5) should be performed in parallel with the SARS-CoV2 peptides. Making experiments with CMV peptides in "other 4 patients at different time points is not a control experiment of the experiment of lack of functionality of SARS-COV-2-T cells done in perhaps (?) other patients.

In any case, I think the experiments now showed in figure 5g and h are quite convincing and can be used to say that there are modifications of the functionality of CD8 T cells during acute and convalescent phase even though the data cannot be used to demonstrate that such functional impairment is only specific for SARS-CoV2 specific T cells. This should be clearly written.

POINT 5.

194. I still think that the data presented in figure 4 that provide a transcriptomic data of the TTV cell of a single patient at single time point when these cells are supposed to be not functional does not provide any important information . The important point could be to perform a transcriptomic analysis of the same TTV-T cells after recover and show what are the changes. The authors wrote that "these results suggest that activated TTD-specific CD8 T cells display a gene expression program of maintained cell survival but restricted T cell (re)activation, proliferation and migration. " Does this fit with the fact that such cells are not functional in the functional experiment?

We thank the Editor and the Reviewers for their comments and input. We have addressed all comments point by point below. All changes have been highlighted in the revised manuscript.

Reviewer #1 (Remarks to the Author):

The authors have addressed the majority of my concerns, with the impressive generation of new experimental data and analysis. I am happy to recommend publication. I do still have concern over the use of "immunodominant" in the title, as I do not think this has been comprehensively established by this study and is a heavily loaded term. I suggest modification.

We thank the reviewer for the appreciation of the new experimental data.

Two major observations strongly suggest the immunodominance of the TTD epitope: 1) the superior magnitude of the TTD-specific CD8 T cell responses compared to other detected SARS-CoV-2 specific CD8 T cell responses both across patients and within patients, which was further emphasized in the comments of reviewer #2, and 2) the high level of TCR diversity based on the TRB-CDR3 sequencing data. However, we understand the reviewers concern and therefore modified the revised manuscript (including the title) by referring to the TTD epitope as an epitope, which has immunodominant features (highlighted in the revised manuscript).

Reviewer #2 (Remarks to the Author):

The authors partially answered my questions:

Point 1: The immunodominance of the TTD-HLA-A01-restricted CD8 T cell epitope is a very interesting point of this paper and the fact that such CD8 T cell response was detected at frequency that appears to be around 10-20% of total CD8 T cells (3 out of 8 patients) and among 1-10% of total CD8 T cells is quite extraordinary (Figure 1C). However, I don't understand why that the authors do not somehow point out that such incredibly high frequency directly ex vivo were never found in SARS-CoV2 infected patients studied so far (it is only mentioned briefly in the discussion). Classical frequency of CD8 T cells during acute disease is always less than 0.5%. This is why I think it will be important to show more control and to test in more HLA-A01+ SARS-CoV2 patients the behavior of such response. The authors partially answer to my question and they add additional 4 HLA-A01+ SARS-Cov2 patients and 2 HLA-A01 healthy controls. It is better than nothing but certainly does not address for example the problem of why such high frequencies of TTD-specific CD8 T cells were detected almost exclusively in critical and severe patients. In reality if we look at the data, the real interesting point is that the authors found extremely high frequency of TTV-HLA-A01-specific CD8 T cells only in severe/critical patients while frequency of this TTD-specific CD8 T cells is very low in the only HLA-A01 asymptomatic HLA-A01+ patient studied. I still think that a larger quantity of HLA-A01+ SARS-CoV-2 infected patient with mild and asymptomatic infection should be studied to better understand whether high frequencies of such TTV-CD8 T cells are present only in severe/critical cases. I also think that a complete set of dot plots with appropriate controls (it should be shown to display such high frequency of TTV-specific CD8 T cells. Controls of the staining of TTV-A01+ CD8 T cells in non-HLA-A01+ severe patients should also be shown to see the level of background staining detected.

Some longitudinal data were also added and they are also very interesting (Figure 5). Frequency (patient 143 and 117) of the TTV-A01-restricted CD8 T cells remains very high (in 117 remains at about 2% of total CD8 T cells after recovery) and in patient 143 remains superior to 1% of total CdD8 T cells. It is not very clear why the authors are describing such data saying that "there is a decrease of the magnitude of such CD8 T cell response". This is certainly true in

patients 096 but it is not true for 117 and the persistence of a CD8 T cell response specific for a single epitope at a frequency higher than 1% (patient 143) is certainly not so common. This is why I still think that the CD8 T cell response against this epitope, that is the single focus of this work should be better analyzed in more subjects.

We thank the reviewer for making this point. We agree that the magnitude of the identified TTD-specific CD8 T cell responses is remarkable. This finding has now been emphasized in the revised manuscript (page 4, line 120 and page 8, line 238). In addition, dot plots of all detected TTD-specific CD8 T cell responses in HLA-A*01:01 positive donors and appropriate controls in HLA-A*01:01 negative donors are now included in the revised manuscript (Fig. S3).

We agree that future studies are warranted to further dissect the pattern of the TTD-specific CD8 T cell responses during acute mild and moderate COVID-19 disease. Unfortunately, we do not have these samples available at this point in time. Patients with mild/moderate disease usually do not require hospitalization and recover at home in quarantine. It would be ideal if there were enough personnel available and logistics in place to collect samples at home, however, this is currently not a possibility.

Of note, the sample analyzed from the one HLA-A*01:01 positive asymptomatic patient was collected 3 months after the positive PCR test for SARS-CoV-2. Unfortunately, no samples were collected from the asymptomatic patients at the time of the positive PCR test as they were at home in quarantine as well.

We thank the reviewer for the comment regarding the longitudinal data and the change of the magnitude of SARS-CoV-2 responses, and we apologize for the lack of clarity. This has been clarified in the revised manuscript (page 7, line 211).

POINT 2: The authors agree than their data cannot be used to generally conclude that ORF-1 is more immunogenic in the totality of the patients. This could happen only in HLA-A01+ subjects. They have modified the text. However, in the abstract they are writing that “A total of 18 SARS-CoV-2 epitopes were identified; 8 of these epitopes were derived from the ORF1ab”. The sentence should be changed since it seems to imply that ORF-1 is in general more immunogenic”

We thank the reviewer for this comment. The abstract has been modified accordingly (page 1, line 8).

POINT 3: The experiments of functionality remain problematic. It is likely that the CD8 T cells analyzed during the acute phase of severe COVID-19 were functional impaired. These data are in line with recent results showing that “functional SARS-CoV-2 specific T cell response is present during acute infection only in subjects with very mild infection and not in severe cases. (Rydyznski Moderbacher C, et al. Antigen-Specific Adaptive Immunity to SARS-CoV-2 in Acute COVID-19 and Associations with Age and Disease Severity. Cell 2020; 183: 996–1012.e19. and Tan AT, et al. Early induction of functional SARS-CoV-2-specific T cells associates with rapid viral clearance and mild disease in COVID-19 patients. Cell Reports 2021; 53: 108728–13.). These data should be quoted, and the results discussed in the context of these new findings while the discussion about T cells are not functional because they are not cross-reactive with other common coronavirus (page 9, line 276-86) is quite implausible (and are we sure that the TTV sequence is not shared among other pathogens?).

We thank the reviewer for this comment. We have included both suggested references in the discussion (page 9, line 289).

We had initially assessed the cross-reactivity of detected SARS-CoV-2-specific CD8 T cell responses based on the comparison between SARS-CoV-2, SARS-CoV-1 and the four characterized coronaviruses causing the ‘common cold’. No overlap with these coronaviruses was found for the TTD epitope (Table S1). To further address the reviewer’s comment, we performed a protein-protein BLAST using the entire NCBI database for the TTD epitope together with a test of the predicted MHC binding affinity and likelihood of cross-recognition of a TTD-specific TCR with other peptides similar to the TTD epitope. We found that the epitopes most similar to the TTD epitope in terms of amino acid characteristics were unlikely to be presented and the chance of a cross-reactive TCR was low. These observations indicate that, based on current available knowledge, the TTD epitope is specific for SARS-CoV-2. However, as not all pathogens are in the database, we have modified the revised manuscript accordingly by using the term ‘specific’ instead of ‘unique’ for SARS-CoV-2 (page 3, line 94).

POINT 4: In addition, the lack of control of such functional experiments remains. To demonstrate that SARS-COV-2-specific CD8 T cells are the only T cells that are not functional during severe COVID-19, the control experiments performed with peptides of other viruses (included Figure R5) should be performed in parallel with the SARS-CoV2 peptides. Making experiments with CMV peptides in “other 4 patients at different time points is not a control experiment of the experiment of lack of functionality of SARS-COV-2-T cells done in perhaps (?) other patients.

In any case, I think the experiments now showed in figure 5g and h are quite convincing and can be used to say that there are modifications of the functionality of CD8 T cells during acute and convalescent phase even though the data cannot be used to demonstrate that such functional impairment is only specific for SARS-CoV2 specific T cells. This should be clearly written.

We appreciate this comment from the reviewer. The requested control experiments could not be performed for more patients due to two major limitations: 1) limited total CD8 T cell counts due to lymphopenia in COVID-19 patients, and 2) low magnitude of detected CD8 T cell responses specific for other viruses and therefore insufficient cell numbers that do not meet the required threshold for the peptide stimulation assay. However, sufficient cell numbers for the requested control were available for patient COVID-131. It is important to note that the magnitude of the CMV-specific CD8 T cell response in this patient was 1.3% and 0.7% of total CD8 T cells based on the pHLA multimers and the IFN γ assay, respectively. In contrast, the TTD-specific CD8 T cell response was 19% and 2.7% of total CD8 T cells based on the pHLA multimers and the IFN γ assay, respectively. The difference in magnitude between the pHLA and peptide stimulation assay was 2-fold for the CMV response and 7-fold and the TTD-specific CD8 T cells response. This observation indicates that the TTD-specific CD8 T cell response is considerably more dysfunctional compared to the CMV-specific CD8 T cell response. This information is now included in the manuscript, even though this data is restricted to one patient, which has been specified (Fig. S4, page 5, line 150).

POINT 5: I still think that the data presented in figure 4 that provide a transcriptomic data of the TTV cell of a single patient at single time point when these cells are supposed to be not functional does not provide any important information. The important point could be to perform a transcriptomic analysis of the same TTV-T cells after recover and show what are the changes. The authors wrote that “these results suggest that activated TTD-specific CD8 T cells display a gene expression program of maintained cell survival but restricted T cell

(re)activation, proliferation and migration. “Does this fit with the fact that such cells are not functional in the functional experiment?”

We thank the reviewer for this comment. We agree that it would have been beautiful to have matching transcriptome data on TTD-specific CD8 T cell responses during acute COVID-19 disease and convalescence. However, the available samples and cell numbers were limited and we prioritized the functional assay to address the reviewer’s comments regarding the longitudinal analysis.

The transcriptomics analysis is now based on two batches of data, and we detect a strong overlap in the gene expression profiles of the TTD-specific CD8 T cells in comparison to bulk naïve and effector/memory CD8 T cells between these two batches. Furthermore, the transcriptomics of the TTD-specific CD8 T cells during acute COVID-19 disease show a gene expression program of genes associated with inhibition of (re)activation which appear to fit well with the lack of ability to produce cytokine in response to peptide stimulation. This has now been clarified in the revised manuscript (page 7, line 204).

REVIEWERS' COMMENTS

Reviewer #2 (Remarks to the Author):

The authors have added some minor modifications. Even though the number of A11 + recovered patients is very small the observation reported is important.